# Performance Evaluation of Land Administration System (LAS) of Nairobi Metropolitan Area, Kenya

**Daniel Orongo Nyangweso [1,2,*] and Mátyás Gede [3]**

1 PhD School of Earth Sciences, ELTE Eötvös Loránd University, Pázmány Péter sétány 1/A,
H-1117 Budapest, Hungary
2 Institute of Geomatics, GIS and Remote Sensing, Dedan Kimathi University of Technology,
Nyeri 10143, Kenya
3 Institute of Cartography and Geoinformatics, ELTE Eötvös Loránd University, Pázmány Péter sétány 1/A,
H-1117 Budapest, Hungary; saman@map.elte.hu
* Correspondence: orongo@map.elte.hu; Tel.: +36-202845693

**Abstract:** This paper aims to evaluate the internal processes of the current land administration in Kenya based on the following parameters that include ownerships, transactions, transfers, inquiries, public records of maps as attributes, issues, and customer satisfaction using stakeholder surveys and focused group discussions. A framework tool was developed for evaluation and shared with potential respondents who were either clients or staff working at the Ministry of Lands to obtain an overview of the performance of the documentation and registration processes of the land administration system (LAS). Data collected were processed and analysed using SPSS 26. To ascertain data reliability, the Cronbach's alpha test was performed, and a coefficient of 0.908 was calculated, which indicated the presence of high internal consistency of the questions and relevance of the study subjects for the participants. The findings revealed the presence of emerging issues where an average of approximately 28% of clients do not have an idea of land registration transactions. In addition, in Kenya, similarly to other national mapping agencies in the developing world, pre-independence laws have begun, which need to be upscaled or revised to sustain and effectively address issues noted on land administration and policy.

**Keywords:** gazetteer; land use; land administration system; land tenure; policy

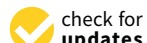

## 1. Introduction

### 1.1. Background

In real property, interest in land is the right to acquire and own land. Interest in land can either be legal or beneficial. Legal interests refer to formally registered interests in land or rights held in land and beneficial interest to acquire benefits from land. The interests in Kenya are governed by the constitution of Kenya, under article 65 [1] and parts 2, 3, 4, 5 and 6 of the Land Act [2] and part 4 of the Land Registration Act [3]. Interests in land and the registers govern proper land administration when documented and processed properly. The lagging registration of interests in rural lands away from the urban zones has made handling rights in those lands difficult. The delay in registration has resulted in non-registration and a lack of verifiable records of rural property to ascertain the authenticity of the information provided by proved occupations. Regulation of overriding interests has become difficult due to the lack of sufficient systems in most African states to verify and validate records due to the lack of continuous performance monitoring of land administration systems. One of the solutions already being applied in developed countries is the use of a cadastral gazetteer that integrates the register with the location of parcels in maps or topographic maps. The integration of the cadastre with a gazetteer is useful in the provision of searches of locations of parcels through the use of documents describing the locations of the registration section names, the topographic maps, the survey plans,

and a digital gazetteer of geographical names. An assessment of geographical names and addresses in EU gazetteers found that an EU gazetteer is needed to support multi-national applications that serve to answer queries on emergency responses; search for records; and search for new datasets, new objects and cultural heritage [4], with most cadastres needing to be maintained and updated and should be widely available as in developed countries. Issues such as access, survey accuracy, seamless cadastres, and online transactions persist, such as the case for the Australia Cadastral system [5,6].

Integrations of the cadastre with land registry or place names gazetteer follow different integration approaches. Some have implemented the geographical place name gazetteer to run side by side with the cadastre or register, while others prefer hosting the cadastre and place name gazetteer running independently but with the provision of links to access resources in the gazetteer and cadastre. In some instances, the cadastre and land registry are integrated for the countries such as Belgium, Bulgaria, Cyprus, Czechia, Estonia, Greece, Croatia, Lithuania, Luxembourg, Montenegro, Netherlands, North Macedonia, Romania, Russian Federation, and Spain, while others have separate cadastre and registry databases [7].

Developed countries such as Austria, Czechia, Denmark, Finland, France, Germany, Greece, Italy, Luxembourg, Netherlands, Poland, Portugal, Spain, Sweden, and the UK have continuously incorporated addressing gazetteers in the cadastres in some sectors [4,6,8–14], often seeking stakeholders' feedback for updates [15] for the identification and location of real estate. Farmers' survey of agroforestry (AF) perspectives in Czechia revealed that areas under AF are less than 1% and that there is no legislation defining the land use system despite high interests in it. Issues affecting the AF land-use system of Czechia include high costs of establishing it, complicated rules in legislation and uncertainties of obtaining returns of investment [13]. In addition, the cadastral system in some EU countries is parcel-based, where some cadastres contain building data [10] or support 3D cadastres due to possibilities in new technologies [12]. In addition to countries such as Bulgaria, Cyprus, Estonia, Hungary, Latvia, Lithuania, Malta, Norway, Poland, Romania, Slovakia, and Slovenia, they have been updating their cadastres with new data using new technologies [4,16–18].

Similarly to Kenya, other countries in Africa have poor cadastres, hence rendering servicing of loans and tax compliance difficult in the real estate sector. In Kenya, the overriding interests include leases of 999 years, converted to 99 years where some are about to expire and revert to the state [19]. Other overriding interests include legal easements, public rights of way for electricity, water pipes, communication cables, legal easements and 'profits prendres' (French for 'right of taking'), non-statutory rights on riparian reserves or embankments, etc., for which their conditions vary from country to country based on applicable existing or new laws enacted to address difficulties of implementations by land administration systems especially in the developing world.

Evaluation of land administration system (LAS) involves measurements of the performance of an organisation to improve efficiency, production, and performance [20]. There is limited literature on the evaluation of land administration, specifically on the internal processes and the role of gazetteers in land administration. Adopting any one method to be unilaterally acceptable for all cases poses a risk of bias since many countries have existing heterogeneous differences in LAS, ranging from differences in the level of development, the pace of enacting of new lands laws, spatial heterogeneity issues in discovery, and addressing issues in the LAS to differences in financial, political, religious [21], technical, and user cases.

### 1.2. Land Administration System (LAS) Performance Evaluation

The first evaluation on a large scale involving the application of the European Foundation for Quality Management (EFQM) was first implemented in Europe in countries such as Poland [22] and the Netherlands [23]. Case studies of its implementation in other areas are available online on the EFQM website [24]. The critical evaluation of the EFQM

found that the two results variables used are not sufficiently correlated with others. The model fails when running a test without the variables [25]. A case study in Ethiopia's Addis urban cadastral system LAS evaluation was performed based on the EFQM [26] framework, which revealed unreliability issues due to issues on the strategic plan, quality of leadership, bureaucratic processes, and supply of resources. Amhara, Ethiopia, while considering external factors, such as monitoring and evaluation functions, reviewed its LAS using literature reviews, interviewing stakeholders using questionnaire surveys, and group discussions [27]. The findings from the identified strengths, weaknesses, opportunities, and threats about the Amhara LAS identified problems with tenure security due to the lack of reforms and poor public participation. Based on advocacy coalition theory, relying on desk review, telephone interviews, and qualitative reviews revealed the feasibility of using two different frameworks and separate instructions [28]. The evaluation of land use/land cover impact using remote sensing in Dhaka, Bangladesh, indicated rapid development of built-up areas and subsequent reduction in urban ecosystem service value [29]. The primary causes of rapid developments were attributed to issues with policy and planning in the LAS of Bangladesh [30]. In Zimbabwe, a review of land property rights, land tenure systems, and periurbanity of Domboshava for four years from 2011, inclusive of site visits in 2019, applied Anthony Gidden's structure agency theory, which states that 'just as structures influence an individual's autonomy, structures are sustained and adapted via the exercise of agency', to analyse data. The study revealed that land transactions do not favour women [31]. The root causes for the issues indicated problems for women in assessing and holding land use due to an unfavourable LAS in Zimbabwe.

Various studies have been conducted on the LAS of Kenya. First, a study that assessed the effectiveness, efficiency, and quality of LAS indicated that LAS performance was below expectations compared to other developing countries [32]. Second, a study using a Land Governance Assessment Framework (LGAF) tool in assessing the LAS of Kenya indicated issues of poor enforcement mechanism of new legislations; that the majority of the land is not registered; the protection of the rights of women and other marginalised groups is not guaranteed; and the presence of restrictions on land ownership in urban areas and processes are slow and duplicative and always involving many institutions [19]. Third, an evaluation considering the improvement of the structure of the LAS regarding the quality of the cadastral maps as per the requirements of a modern LAS established that there is a need to consider special characteristics of maps when integrating them with LAS [33]. The fourth evaluation of Kenya's LAS, using the multi-value vector maps approach and Smith's normalisation procedures, explored methods of modernising it and revealed that the administrative nature of LAS is bureaucratic, complex, duplicative, and slow [34,35]. Fifth, a study investigating the possibilities of mapping unrecorded land rights found that SmartSkeMa and UAV can update the LAS map database of Kenya [36]. The present study is different in that there are no studies on the performance evaluation of LAS in Kenya, particularly those focused on processes. Secondly, the studies were performed when the new land laws addressing the issues noted for each study were different, as reflected in their recommendations.

Notwithstanding the existing literature on performance evaluation, there is scant scholarship on the performance evaluation of land administration focused on internal processes of an organisation in Kenya or globally. There are various performance measurement approaches for evaluating an organisation's performance, which an organisation mostly uses to upscale performance [37]. They include the Balance Score Card (BSC) [38] and the European Foundation for Quality Management (EFQM) [23–25,39], which provides insights for organisational to change and improve performance and works together with ISO 26000 [40] for sustained performance pegged on social and environmental perspectives as per the Sustainable Development Goals (SDG) of agenda 2030. However, in its criteria, it provides contribution of processes to the 10% evaluation and 10% stakeholders. In addition, the Workflow Management Coalition (WfMC) standards provide processes to be followed

for software products as well as a Statistical Process Control (SPC) [41,42], and it applies statistical tools to monitor and control processes in order to obtain returns on investment.

Similarly, although suited for institutions offering training, the ISO 9000 certifications 2015 [43] also measures the quality methods to be certified for institutions through a training process to realise goals. A different approach for evaluation is the Integrated Performance Measurement System (IPMS) [44] as a method to evaluate processes used by organisations in their operations [44]. As per Wibisono's theory, internally developed solutions are key for bearing returns of investments in response to the deployment of appropriate innovative technologies. The technologies include global positioning systems (GPSs), UAV, continuous observation systems (CORs), geographic information systems (GISs), and use of information and communication technologies (ICT) that serve as part of innovative methods to address issues on LASs.

Principles of policy, tenure, administration and cadastral; institutional structures and infrastructure; ICT solutions; and human resources as indicators and aspects are key parameters of performance evaluation [20]. In addition, each of these evaluation methods focuses on entire organisation units rather than the process. Measuring the definitive quality and quantity aspects of performing a measurement should focus on the process and the actors or stakeholders in the process [45]. Each of the above methods has pros and cons regarding applicability since there is no approved evaluation standard, and any can be used. In this research, the author focuses on the internal process of the Integrated Performance Measurement System (IPMS) method of evaluating land administration as a process based on the internal process of an organisation to measure performance [44].

In detail, the IPMS proposal of Wibisono (see Table 1) involves measuring a company's vision, mission, and strategy by using perspectives of organisational output, internal process, and resources capabilities to support decision making. The organisation results include financial and non-financial aspects where financial ratios are used to evaluate the current business situation. The financial ratios used include profitability, activity, and liquidity ratios. Non-financial aspects include consideration of stakeholders of the organisation such as customers, suppliers, employees, and the government.

**Table 1.** Perspectives of an Integrated Performance Management system (IPMS).

| Perspective | Aspect |
|---|---|
| Organisation results | Financial |
| | Non-Financial |
| Internal process | Innovations |
| | Operation process |
| | Marketing |
| | After-sales service |
| Resource's availability | Human resources |
| | Technology resources |
| | Organisation resources |

Adopted Source [44].

The internal process is the organisation's operations activities that affect business outputs such as innovations established and operation processes involving a specified number of clients while fulfilling its mandate and achievement of yearly targets to maintain serving the clients. At the same time, the resource's availability involves the capabilities of the organisation, human resources, and technology (see Table 1).

The present studies evaluated only the internal process of the Kenya Land Administration System (LAS), focusing on the Nairobi Metropolitan area. The performance evaluation of the internal processes using the approach of [44] was chosen due to its simplicity in usage; having stakeholders on board; using fair evaluations; and providing accountability and responsibility to all actors [46]. Furthermore, in its method, one can set objectives and

performance indicators for internal processes based on the aspects of the methodology for a specific organisation to be evaluated.

Since the land has three attributes to be assessed: land ownership, land value and land use [47], and most of the scholarship deals with either land use or value, the contribution of this paper is to add scholarship addressing the evaluation of an LAS processes by using the IPMS [44] approach and stakeholder surveys [48] to evaluate the process used in the transaction within Kenya's LAS by relying on stakeholders for feedback on the new laws enacted.

The United Nations Economic Commission for Europe (UN-ECE) guidelines define a land administration system, in its basic form, as being able to "determine or adjudicate land attributes, record and disseminate information on the tenure, value, and use of land when implementing management policies" and to ensure the security of tenure [49]. The sample guidelines of UN-ECE are not the universally required or accepted standard to be used because of heterogeneous differences among countries in economies, procedures, and application of advanced LAS. However, in introducing a new LAS, where land registration, cadastral surveying, and mapping are the main processes, the guidelines serve as a guide. The guidelines define legislation, databases, organisation, and maps as the key elements required for a good LAS. As in other African countries, land in Kenya was reserved for men only under the laws and was only transferred or inherited by men; thus, women had no equal rights to inherit ancestral land and marital property [50]. The Matrimonial Property Act [50] replaced the Married Women's Property Act, 1882, of the UK [51]. The new constitution [51] enaction into law paved the way for women to own land as property. However, despite the new laws being in place, the struggle to own land for women and marginalised groups continue unabated [31].

Based on classical cartographic themes, each land use requires diverse data attributes for real estate construction purposes. The attributes range from authoritative ownership data, parcel bounds and the rights held within physical planning regulations. The authoritative cartographic data and processes followed are necessary for obtaining approval from various related stakeholder agencies. Authoritative geospatial data are data that are preowned and authored by authoritative agencies for use in registering land ownership. Marks of symbols and text are used to define the authoritative cartographic representations of land, its boundaries, and its description. Authoritative cartographic data aid in the interpretation of individual properties. It includes representation of beacons and their associated symbology, area sizes, parcel numbering, bracing (to identify whole and part relations) and abuttals in describing the location of a parcel of interest.

The problem for most real estate developers is obtaining the necessary authoritative attribute data. It ranges from general ownership inquiry, land sizes, location, description, and user status approvals. The research evaluates the internal process aspects of innovative technologies employed in LAS including its operation processes, documentation, and after-service satisfaction feedback as one of the three methods [44] for assessing the internal process performance in land registration. The performance indicator for each attribute aspect includes the Ardhisasa online system [52], the inquiries made, documents submitted and requested and the overall after-service client feedback. The assessment uses schematic data attributes and their relationship with user needs during the inquiries. Ideally, the research attempts to answer three questions: (1) what users are interested in during inquiry on real estate properties; (2) whether stakeholders are aware of the process of transactions and the services available in the Ardhisasa online LAS or other locations in respective counties; and (3) what are the required documents for making registry inquiries or transactions, under the new laws, in addition to the role of gazetteers in cadastral boundaries. The research outcome will aid in understanding the emerging issues for all stakeholders in the land sector where transactions on land occur under the current LAS dispensation.

An assumption is made that lacking necessary land attributes for registering land documents at different stages of land registration shapes people's experiences in a land transaction in authorities involved in land in their community and in society to address the

research questions. The experiences of the registration process can correlate positively or negatively for successful applications based on the attributes used in registration, the person performing registration, and the nature of the transaction from the start to the end. In addition, a literature review was performed on registration systems to identify boundaries of properties with views of incorporation of the use of GIS technologies using traditional approaches where, in most cases, paper maps or survey plans are used [53]. However, there are no systems in others; hence, the transaction processes in land administration and governance on the launched digital LAS still use paper-based document record systems. Another issue is that interest land is ever incremental, and the registers need proper land governance.

A review of LAS in sub-Saharan countries where new laws have been enacted indicates that land policies and new laws must be adopted and implemented after evaluating and monitoring the process. The approach may assist in addressing challenges of land questions and access to the sustainable usage of land [54] to realise sustainable development goals (SDGs).

The enaction of new laws has resulted in additional challenges occasioned by the introduction of English or European colonial laws to manage rural lands since the start of colonialism, during colonialism and after independence [54]. Customary-based rural lands heavily rely on trust and mutual ownership based on groups of people or communities, which primarily rely on communal land tenure as potential proponents for the Torrens land administration. The Torrens system started in Australia, and its application was introduced to Canada, the Dominican Republic, Israel, Malaysia, New Zealand, Philippines, Singapore, Thailand, the UK, and several African countries, including Kenya. The Torrens system, loosely known as the 'deed system', operates on three principles: mirror, curtain, and indemnity, which still applies to the coastal areas and several urban areas, until all land conversion to the new registered Land Act 2012 is completed.

Furthermore, concealed cadastral boundary records also pose problems in delineating the land through zoning and development control processes and end with the actual showing to a beneficiary. Place names are suggested to be known to project designers, engineers, or project owners and are ultimately chosen as area place names. The name proposed or indicated on the developed scheme plan is verified and authenticated at survey offices, but the real definition of the narrative description where the parcel or border is located is not included. Second, in some instances, the source of distinct cadastral names comes from a land-buying firm based 100 kilometres distant that buys a property and transfers the name to that new parcel. The name is then historically coined to those properties transacted by the land-buying company. Third, typical renaming practices, while exceptional, may entirely confound even cadastral border names and completely identify the actual place, particularly through the branding of names or generalisation bias. The issue of protracted high prices for land arises when the property value is likewise applied as the base value for new sites with no economic effect. Therefore, confusing the real estate market by bringing artificially high property values known to exist in the estate businesses' original locations but brought to a new area. As a result, if all place names in a gazetteer are given name connections to all characteristics they represent or specify, the problem will be identified and mitigated as defined in ISO 19112:2019 [55]. ISO 19112:2019 acknowledges that 'a gazetteer is a subtype of a register' as defined in ISO 19135 [56], and that land and location must be specified as 'location'. Class refers to an item class, and location refers to a 'register item'. A cadastral gazetteer is described as a place or position accompanied by a description, which can be a label, code, or coordinate tuple as per spatial referencing criteria for cadastral gazetteers with street or road adjacency ID or name. The use of a cadastral gazetteer is anticipated when actors in the real estate sector offer inadequate data and the public rushes to invest in real estate projects with no prior knowledge of their location, amenities nearby, and closeness to transport facilities.

Issues of poor revenue collection on land, corruption, a poor filing system, excessive administrative expenses, nonself-checking of duplicate documents, and delayed search

processing plague most LASs of most nations since most are not linked as a cadastral gazetteer owing to too many players and restrictions. Stakeholders of LAS in Kenya include financial institutions, the Ministry of Lands and Physical Planning, Ministry of Roads, water resource management authorities, electoral bodies, wildlife service authorities, individual and group proprietors, National Land commissions, Institution of Surveyors, Law Societies, planners, forest service authorities, national environmental authorities, revenue authorities, and the Land Control Boards, all of which rely on a single register record for each transaction. Each actor plays a significant role in processing the document resulting in the transfer and registration of new land ownership. Recently, in Kenya, a digital system was launched to manage the data online, using the Ardhisasa National Land Information Management System (NLIMS) platform [52], to manage, document, and hasten the processing of documents to address the concerns in the current LAS.

To address the emerging issues such as the comprehensive documentation of land objects, including rights, restrictions, and responsibilities of owners and stakeholders, a land administration domain model (LADM) [57] has been proposed as an International Standardization Organization ISO 19152: 2012 [58] model implemented by the International Federation of Surveyors (FIG). However, LADM has been deemed impractical in most African and developing countries due to varying social tenures for individuals or groups of communities prompting for the development of the social tenure domain model (STDM) based on governing and addressing rights and claims for the singly owned parcels thereof, after improving it for SDGs achievements [59]. Open-source land administration software was developed based on ISO 19152, a reference for LADM. Other proposals complement authoritative data with volunteer data frameworks [60,61]. The use of a fit-for-purpose (FFP) corroborative framework is poised to address some issues noted in STDM, for which their aspects have assessed issues pointed out in FFP LAS, and some authors proposed the use of top-down and bottom-up approaches [62,63].

### 1.3. County Boundaries and Place Names in Kenya

Toponyms as used describe administrative divisions and localities in Kenya where boundaries are defined using registration section names, place names, and lines. Each county has defined boundaries maintained and demarcated by the Independent Electoral Boundaries Commission (IEBC). However, there has been a discrepancy in the place names described and kept on the ground and governed, prompting numerous cases to inform litigations for redress. The litigations are self-pitying for counties since the problem may be due to changing geographical names with static boundaries and geopolitics, which affect place names. The boundary conflicts can be indicated by the disputed county boundaries pitting Kisumu and Kakamega for the location of Maseno and whether it should be renamed Siriba or maintain the status quo. Additionally, there are the cases of Isiolo and Meru, petition 515 of 2015 [64]; and the case of Turkana County and West Pokot and Baringo Counties petition no 113 of 2015, seeking territorial integrity [65].

In addition, the cases of Machakos and Makueni claim of Konza ranch, formerly Malili, each claiming ownership of the range, are only a few cases that highlight the problems emanating from lack of clear interpretation of descriptions in boundaries prompting legal interpretations from the courts. The problem arises mainly when the descriptions in the text may have changed or been altered to suit interpretations of the boundaries. Hence, the only remedy is to use cadastral boundary names. By default, the boundary names of cadastral boundaries still maintain their integrity of correctness even if place names change. The uncertainty is that, for one to make a better model evaluate the influence, one must evaluate the records found in the cadastral map together with revisions of subdivisions and the associated place name descriptions found on them that have no historical proof.

### 1.4. Land Administration and Management Policies

Various scholarships exist for the administration and management of land. The World Bank Group statistics on the land portal foundation website [66] document issues on land

conflicts. In the land portal, land governance studies have revealed country-specific trends where various issues have emerged on how each country manages and the strides made to tackle them.

African countries, including Kenya, have low land tenure securities, where 32 countries from Africa dominate those with low security of tenure of less than 70%, where only Ethiopia, Malawi, Mauritania, Tunisia, Senegal, Algeria, and Egypt have tenure security between 70% to 80%.

The colonisation and annexation of African lands affected countries and territorial, ethnic groupings. Many inhabitants faced forced exclusion from participation and forced movements to guarantee new knowledge of administering and managing occupied lands. There was a notable collaboration of local leaders with the settler farmers or business people in using local expertise in land matters for various indigenous communities to better grip the land resource for fruitful gainful investment. The existing continued relation and connection of the community's dependency on land as a resource benefiting local communities must be acknowledged on how they administered land in addition to impacts of the new technologies [36,67]. The new technologies include Global Navigation Satellite Systems (GNSS), SmartSkeMa [68] (a system for documenting formal and informal land tenures), and unmanned aerial vehicles (UAVs) in geomatics (somehow used to the disadvantage of local communities) to address land issues noted by land questions all over the world due to their cost, unproductive land tenure systems, and ethnic clashes.

Sustainable usage and conservation of land resources help hasten and achieve most SDGs [69], in addition to local and national initiatives launched online to address issues on land from a global perspective. The Kenya land alliance [70] Non-Governmental Organization (NGO) was formed in 2013 to strengthen the community and women's land and property rights, root for better land governance and act as an agent for the marginalised groups affected by land issues. Kenya's land policy of 2007 [71] provides key measures to be undertaken to address issues in land administration. The policy defines land in Kenya as public, community, or private and promotes productive and sustainable conservation.

The World Bank statistics of land tenure security in 2020 [66] for 139 countries provide insights into the issues pertinent to each country regarding land administration and management. Finland ranks as the best country with the highest security of tenure as per the 2020 statistics of 94.39%, Austria 93.55%, and Sweden 92.34%. Additionally, Kuwait, 47%, tops the list for the countries with the least security of tenure, followed by the Philippines 51% and Liberia 51.12%. Kenya ranks at position 24 with 60.74% security of tenure and women have 3.4% land ownership in Kenya as per the 2014 World Bank statistics [72]. Issues noted for Kenya include tenure insecurity, forced eviction, inequality in land distribution, and corruption or land grabbing [73], which resonates with the issues noted in the LAS of Tanzania [74]. In Kenya, laws governing LAS have been enacted, such as the National land policy, approved in 2009. The land policy approval paved the way for enacting new land laws such as 'The Land Act 2012', 'The Land Registration Act, 2012', 'The National Land Commission Act, 2012', 'The Environment and Land Court Act, 2011', 'The Urban Areas and Cities Act, 2011', 'Matrimonial property Act, 2013' [50], and 'The Constitution of Kenya 2010'. These laws can be used to address some of the issues noted in previous statutes, as noted by [73].

This paper examines the land administrative paradigm, based on a review of the typical land evaluation process's insights using the IPMS framework method, based on the process from users' and actors' perspectives using questionnaires distributed within the Nairobi metropolitan area [75,76] (see Appendix A Table A3). It is organised into five sections. The first section introduces land concepts such as policies, land registration statutes, and some terms related to land transactions, specifically land registration. It further explores the role of gazetteers in land administration and a typical example of cases pitying county boundaries and place names and the land administration and management policies. The second section covers materials and methods. The third section covers the results on demographics of respondents, the assessment made on transactions, documentation, and

land transaction of the LAS of Kenya. In addition, it reviews the nature of detail inquired, documents for making inquiries, type of schematic data requests and land tenure holdings. The fourth section provides insights into emerging issues related to land registrations, measurements of the performance of the online 'Ardhisasa' client interactions and the satisfaction level of clients with the current LAS. Furthermore, a comparison of landownership with gender and inquiries made with knowledge of transactions is also highlighted. The fifth section discusses the evaluation findings of Kenya's LAS by specifically targeting the internal processes' operations and conclusions.

## 2. Materials and Method

### 2.1. Study Area

Nairobi metropolitan has a population of 9,354,580 [77] residents spread in four counties. The area was selected because most of the registration of title's activities occur in the counties within the metropolitan region where most of the surveys are affected by the new laws. In addition, it is a fair representative application for the entirety of Kenya for general boundaries and fixed surveys processes for all major towns. The metropolitan region comprises counties of the Nairobi metropolitan region of Kenya comprising four counties of Nairobi City, Kiambu, Machakos, and Kajiado (see Figure 1a,b) comprising 707,569, 6016, and 21,783 square kilometres, respectively [78].

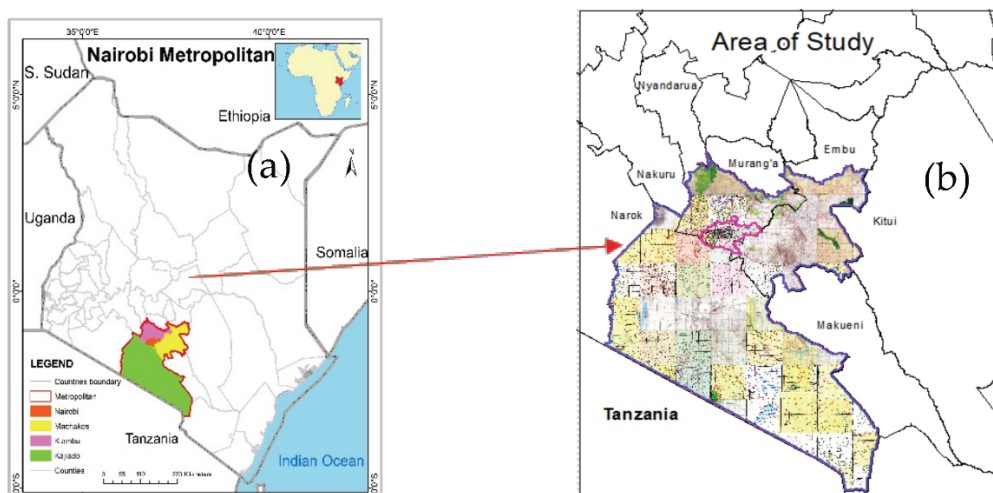

**Figure 1.** (**a**) Location of Nairobi Metropolitan with respect to Kenya; (**b**) location of Nairobi City County with respect to the study area of Nairobi Metropolitan.

### 2.2. Data Collection

Data collection was performed in the metropolitan area.

The questionnaire framework tool served as the primary data source. It was supplemented by secondary data composed of survey plans, topo-cadastral maps, journals, case laws, books, selected survey plans, and topographic maps. Focused group interviews were conducted on the land's business premises, where land office staff assisted in collecting additional data on land registration transactions by conducting discussions to validate the results obtained from questionnaires.

### 2.3. The Study Designs

2.3.1. Research Tools

Overlay data were prepared in GIS software and checked for completeness, attribute accuracy, and granularity. Other tasks performed include physical counts, maps, survey plans, georeferencing, and digitising for overlay and comparison for documenting property attribute validity tests during the group discussions.

### 2.3.2. Questionnaire Distribution and the Stakeholder Survey Sample Size Estimation

The sample size was determined using Cochran's Formula [79] with 95% confidence with a margin of error of 5%. Given that the z score at 95% confidence level is 1.96, the minimum required number of respondents, *n*, is given by the following equation:

$$\text{Ideal sample size, } n = \frac{z^2 \times p(1-p)}{\varepsilon^2} \tag{1}$$

where
*z* is the z score;
$\varepsilon$ is the margin of error or level of precision;
*n* is the ideal sample size;
*p* is the estimated population of an attribute present in the population. Mostly, it is taken at 0.5. By substituting the values in Equation (1), we have the minimum sample size of responses required as follows.

$$n = \frac{(1.96^2 \times 0.5(1-0.5))}{0.05^2} = 384.16 \cong 385$$

Here, *n* = 385 is the minimum number of respondents, and we obtained sufficient data from 401 respondents. However, our population is known; hence, as per the Yamane approach [80], the minimum sample size, *n*, required is given by the following:

The minimum sample size for known population,

$$n = \frac{N}{(1 + N(e^2))} \tag{2}$$

where
*N* represents the population size, and
'*e*' represents the error margin. At a 95% confidence level with a known sample size *n* of 9,354,580 known as per 2019 census [77], a margin of error of *e* = 0.05 is obtained. Then, by substituting the values given in Equation (2), we have the following:

$$\text{The sample size required, } n = \frac{9,354,580}{(1 + 9,354,580(0.05)^2)} = 399.98 \cong 400$$

which yields 400 as the minimum required responses. Morover, 401 respondents participated, affirming that data were sufficient.

The questionnaire was the primary source of data collection tool, which was shared with targeted respondents of officers, staff, and clients of the Ministry of Lands and Physical Planning as actors in the real estate industry. Stakeholders ranged from various categories of professionals who identified their area of the profession when filling the questionnaire, verbally or on their own mobile devices, using Google forms.

### 2.3.3. The Question Items

Questions were designed and shared online and during interviews for potential respondents to fill during their visit at the land's offices to seek service between the 15 October 2021 to 15 November 2021. The respondent's extent area was limited to Nairobi metropolitan areas. The questionnaire contained 14 open and cross-ended questions (see Table A1) administered to selected participants.

The respondents filled out the questionnaires to record their views on land, ownership, transactions, transfer of land, inquiries, land administration process, land registry attributes, land tenure, and other issues. Another questionnaire was completed during the land registry visits. Two additional questions were added to validate the data from questionnaires during the group discussions to assess their level of satisfaction and effectiveness of the current LAS process. For two questions, the Likert scale was used within the

limits of 1–5. In the satisfaction scale, very dissatisfied, not satisfied, neutral, satisfied, and very satisfied were used, and in the group discussion, effectiveness was used as a variable.

The Cronbach's alpha test was performed to ascertain the data reliability of the results and sample size. A coefficient of 0.908 was calculated, indicating the presence of high internal consistency of the items (questions) and relevance of the study subjects for the study subjects.

A factor analysis was conducted to determine the correlation of the variables and the reliability of the results and sample size. Table A5 shows that sampling was extremely excellent (0.900). There was a significant association between variables (high Chi-square value of 6803.391 at 1485 degrees of freedom with a significance level less than 0.001).

2.3.4. Statistical Evaluation of Inquiries of Ownership Details and Internal Processes

Evaluation of the internal processes of the LAS of Kenya involved assessment of innovation, operations used in LAS, marketing, and after-sales service of the internal processes of the LAS of Kenya based on Wibisono's IPMS approach.

The innovation aspect involved the evaluation of the Ardhisasa-LADMS, records, and technologies as indicated by questionnaire item number 13. Operation aspects involved the evaluation of documentation and processes of handling ownership, records, transactions, tenure holdings, inquires, emerging issues, and transfers in the land as indicated by questionnaire items 2, 3, 4, 5, 6, 7, 9, and 10. Marketing assessment involved assessing types of services sought by clients and the map products or documents with schematic attributes as indicated by questions 8, 11, and 13. The after-sales aspect was measured by using a customer satisfaction index using the satisfaction and effectiveness of the current LAS after enacting new land laws and deploying an online service to manage land records. A respective questionnaire item measured each aspect (see Table 2).

**Table 2.** Perspectives of the integrated performance management framework tool.

| Questionnaire Item | Aspect | Parameters |
|---|---|---|
| 13 | Innovations | Ardhisasa NLIMS |
| 2, 3, 4, 5, 6, 7, 9, 10 | Operations | Documentation and processes of handling ownership records, transactions, tenure holdings, inquiries, land issues, and transfers. |
| 8, 11, 13 | Marketing | Type of services offered or accessed and land schematic attributes on various map products or documents. |
| 1,12, 14 | Customer satisfaction | Satisfaction and effectiveness of LAS after implementation of new land laws. |

The parameters used to check ownership details used in making inquiries include land ownerships, transactions, transfers, inquiries, records as attributes, issues noted, validated data or those mentioned by respondents, and customer satisfaction feedback.

During the discussions, respondents also reported new problems regarding land ownership in succession, which also emerged from response data. Cases of land not being transferred are common due to parents dying early without leaving or transferring property to their legal beneficiaries, thus increasing the number of cases where succession registrations are required. There are also reports of children appropriating titles and fraudulently transferring them to themselves. In addition, respondents pointed out that there are increased cases reported in Kajiado. Rift Valley region registers children gifting land to themselves or even organising the murder of their parents before allocating land to themselves through succession or normal transfer [81]. Similar incidents have been reported in the Coastal, Nyanza, and Central regions. Internationally, the vice is not isolated to Africa, and it is also common in Nepal and South Africa. However, succession is performed if the land is not formally transferred to the next of kin in most circumstances. Misinformation has also contributed to an increased number of unsuccessful petitions

killings as land concerns tied to witchcraft, albeit unproven, have persisted due to poverty and animosity among some groups, as resonated by scholarly works and media news [81].

## 3. Results

### *3.1. Demographics of the Respondents*

#### 3.1.1. Gender Representation

In the metropolitan region, the population is 9,354,580 residents composed of 4,647,403 males and 4,706,725 females. The research results indicate that 76.3% of the respondents were males while 23.7% were female (see Table A1). The findings contrast the gender structure of Kenya's population, where the male is 49.7% and female is 50.3%. The population of Kenya has been increasing from 8.6 million as of 1962 to 47.9 million in 2019 [77,82]. The differences may be attributed to disparities and inequalities in the access to land ownership and minimal participation of women in land conveyancing processes. During the survey, all respondents had visited land office registries and survey offices in the metropolitan region or were interviewed either online or in-person and hailed from the metropolitan region.

#### 3.1.2. Age Groups

Overall, 99.7% of the population was drawn from 18 to 59 years and found involved in land matters (Table A2), an age bracket that is more active within a population. The research targeted visitors of any age visiting the land's offices, a time when COVID-19 had started rampaging on the masses and the government had required people to practice social distancing; hence, some disparities on data may arise due to limiting physical contact with participants, especially those that are aged.

#### 3.1.3. Property Ownership

The research looked at property land ownership among public individuals who were active in land administration or management. Findings indicated that 269 (67.1%) respondents own land while 132 (32.9%) do not from a sample of 401.

#### 3.1.4. Land Transaction or Registration Process

Most clients who sought services at lands offices who responded to the survey claimed that they had been engaged in land registration or transfer, with 290 (72.3%) saying they had, and 111 (27.7%) saying they had not (*n* = 401).

### *3.2. Awareness of the Process of Transactions*

Awareness of the process of transferring or registering land in Kenya

The first goal of this research was to find out what stakeholders in land administration and management look for as information on land registration and ownership transfers. Most clients who sought services at lands offices and responded to the survey claimed that they know the entire process in land registration or transfer, with 344 (85.8%) respondents saying they had, while 57 (14.2%) disaffirmed this (*n* = 401).

#### Kind of Inquiries

First, respondents were asked what documents are needed to begin the land transfer and registration process during submission to the registries to avoid unnecessary delays in processing registrations for transferred parcels. However, clients still raise several issues, ranging from land registration and transfer to other land resource management issues. Other issues may also influence response data, such as the fact that most women are not involved in land matters since, before the new land rights regime for women in Kenya and similarly to the rest of the world, men usually owned land. Significantly, most respondents who refused to answer said they did not know anything about land transfers as their husbands usually handled these. Moreover, there was a lacuna in the old constitution and the instruments regulating land alienation. The requirement of spousal consent approval

was introduced to address the issue of land transfers where men transferred land without the knowledge of their wives or when the rights of children were violated by alienating land when disputes arose due to differences in marriage.

Land registration is one of the main functions of the Ministry of Lands and Physical Planning that is problematic to undertake as a core service for delivery. Some issues have resulted in the revision of applicable laws and notices to address emerging issues in the conversion of land titles and block boundaries of the Nairobi Land Registration Unit [83].

Search requests are the most requested services accounting for 72.3% of respondents, followed by inquiry on how to acquire ownership documents (66.1%), then transfers (55.4%), paying of stamp duty (38.9%), succession (34.9%) and placing of caution (17.7%) in that order (see Figure 2). Other inquiries accounted for 5.2%, including questions on registration of charges and discharges, cadastral survey, registration of leases, registration of mortgages, severed land, joint tenancies, conversion of state land to private land, correction of information in title/register, replacement of lost/mutilated title, etc.

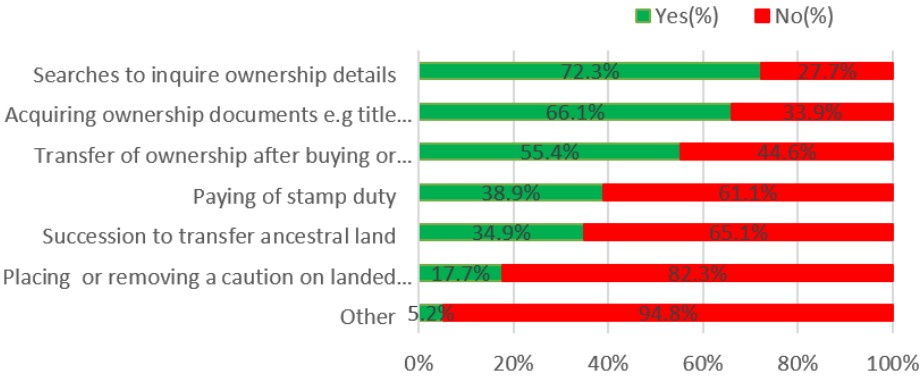

**Figure 2.** Kinds of inquiries at the land's offices (*n* = 401).

Furthermore, others were inquiring on encumbrances attached to land; registration of leases; registration or mortgages; sectional properties; joint tenancies/tenancies in common; conversion of government land to private; correction of particulars on title/register; and replacement of lost/mutilated title.

*3.3. Nature of Details*

Type of details on the plot of land of interest.

Ownership details accounted for most (75.6%) of the attribute data requested on land, followed by the location of a plot (60.6%), area of the plot (48.4%), demarcation of boundaries (40.4%), general inquiries (39.7%), and user-status (28.2%) (see Figure 3). The details sought indicate that leveraging services is needed to switch most of the services to be offered through the digital platform by using a gazetteer linked to it or in the Ardhisasa NLIMS platform, which currently supports only a few services.

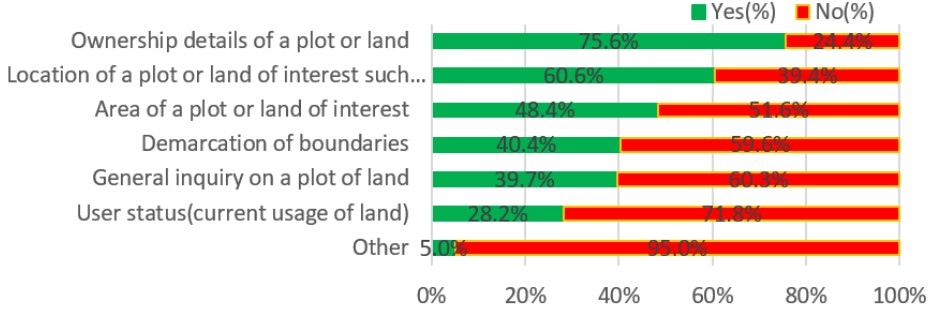

**Figure 3.** Type of details on the plot of land of interest (*n* = 401).

Other details that clients or visitors to the land office may ask about include tenure and transfers; registration of encumbrances and discharges; whether a title deed has been used to obtain a loan or own court orders; cautions affixed to the property and transfer history; other priority rights such as right of way for pipelines and electricity, etc.; encumbrances; the history of the property (green map); file documents and the history of the property and survey, i.e., all previous owners and surveys.

### 3.4. Document Required from Clients for Making Inquiries or Transaction

Documents or particulars used in making inquiries depend on the nature of the inquiry.

The national identity card was the most requested document when seeking government services at land registries, with 71.6% responses. At the same time, the share certificate was the least requested document, with 10.5% of the respondents (private land buying companies issue share certificates) (see Figure 4). Other documents requested by clients include title deed plans (66.8%); transfer forms and sale agreements (51.1%); consent or authority to transfer (42.6%); passport size photographs (39.9%); mutation (25.2%), spousal consent (23.2%); and deed plan (14.2%). Additional documents requested by clients at the time of transfer include documents evidencing consent authority to subdivide in corporate transactions: company resolutions; for succession/transfer—death certificate; grant of administration/probate; confirmation of grant; Kenya Gazette, etc.; rates and rent clearance certificates; stamp duty payment receipts; plot number or parcel number; succession from court and search certificate.

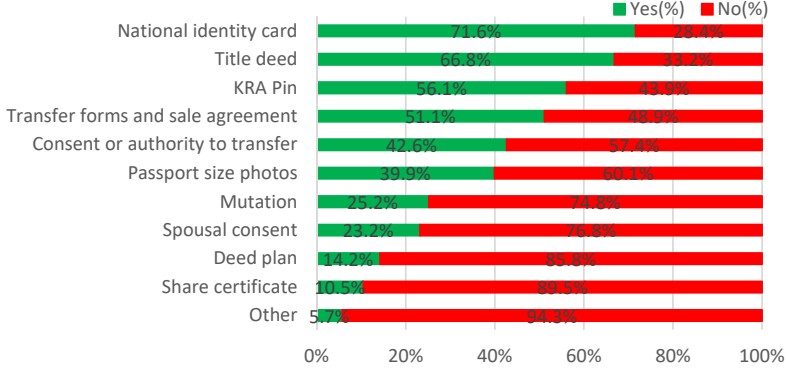

**Figure 4.** Documentation needed to make inquiries (*n* = 401).

### 3.5. Schematic Data

Schematic data attributes used in this context are used for inquiries in property ownerships for location identification.

The most requested schematic data are property ownership data (77.3%), and thematic map data (8%) are the least requested. Others are survey plans (52.6%), mutations (44.6%), control points (26.2%), and topographic maps (19.7%) (see Figure 5). Figure 5 shows that leveraging services is required to move most of the products offered online using a gazetteer linked to the digital Ardhisasa NLIMS platform [52]. NLIMS supports only a few services, but plans are underway to expand services [52].

Schematic map data mentioned by respondents include an encumbrance document, a letter from the bank or land registry stating that a plot of land is free from the credit, the applicable land use policy, the current street and road map, the muster roll, land ownership and applicable regulation, and official research.

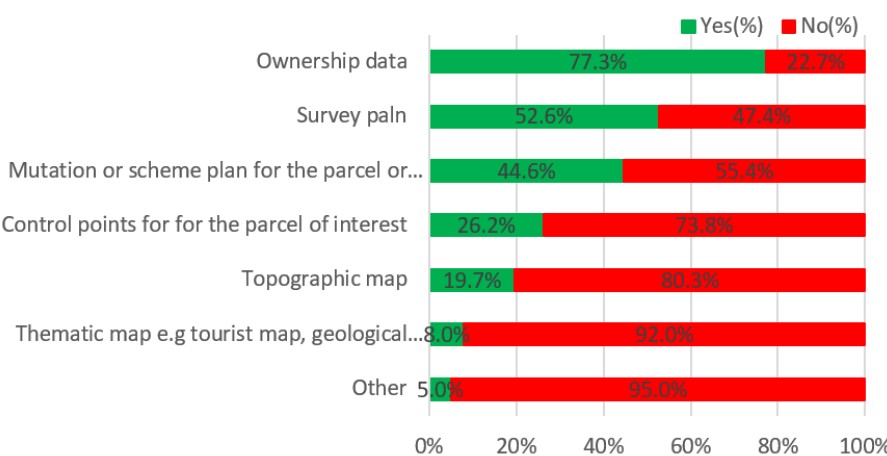

**Figure 5.** Schematic data requests while making inquiries (*n* = 401).

### *3.6. Land Tenure*

Land tenure holdings earmarked for the study included sole proprietorship or group ownerships for privately owned land, tenancy, leasehold (including individual and informal leases), inheritance, and squatting.

Three types of land tenure were studied where 72.1% of the respondents were sole owners or groups, followed by inheritance from parents (41.1%), while 22.7% were tenants or leaseholders and 3% were squatters (see Figure 6).

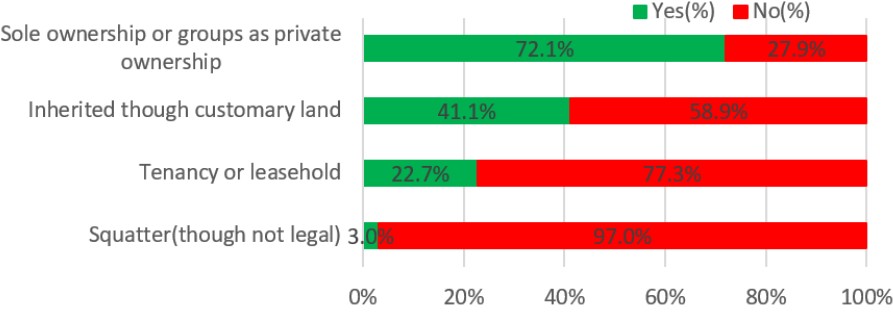

**Figure 6.** Land tenure holdings.

## 4. Issues in Land Registration in Kenya

### *4.1. Knowledge on Issues in Transferring and Registration of Land*

The study's second goal was to find out the causes of problems in registration and emerging issues in land administration to mitigate possibilities of deploying volunteered geographic information (VGI) based gazetteer data on supplementing the publicly available data attributes.

The most frequently cited problem related to land transfers and land tenure is land grabbing (77.1%), followed by double allocation (50.9%), poor filing system (50.1%), absentee landowners (41.4%), illegal land conversion (35.4%), untitled lands (35.4%), squatting (31.2%), exchanged land (23.7%), sale of government land (21.2%), ethnic conflict (16%), and compulsory land acquisition (8.7%), in that order (see Figure 7). Other problems cited by respondents included delays in issuing title deeds for ancestral land; private freehold leases in the coastal region for houses without land; huge tracts of land owned by foreigners on land claimed by local communities in Laikipia and Kwale regions or absentee landowners; numerous squatters; and unregistered lands, all of which emanate from irregular land allocation, consistent with previous research [84,85]. The problem also occurs globally and at the country level [86].

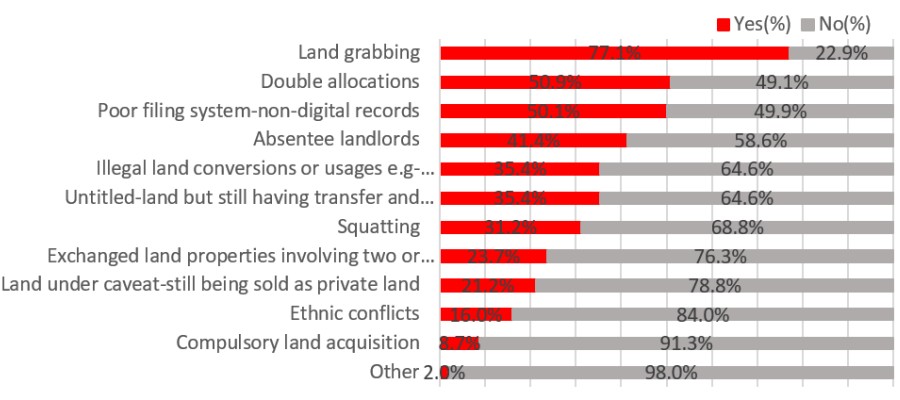

**Figure 7.** Emerging issues on land ownership.

Other respondents indicated that unprofessional surveys are approved, resulting in many boundary disputes; a great majority of land is in the name of deceased owners whose beneficiaries (without legal capacity granted after inheritance) nevertheless sell the land. Some actors (including registrars) generally disregard spousal consent requirements, which respondents believe is due to a lack of public education on land issues.

*4.2. Measurement of the Interactions of Ardhisasa Platform Service*

Measurements of the engagement with the online Ardhisasa platform were investigated to determine the nature of services accessed.

Land registration is the most requested service online with 69.6% of respondents, followed by general inquiries (42.4%), land surveying (42.4%), land valuation (30.7%), physical planning (22.7%), and ICT (10.0%) in that order. Other services on the platform included asking for assistance with formal land searches (see Figure 8). However, some respondents pointed out a lack of public education on land issues and the resolution of boundary disputes through the platform.

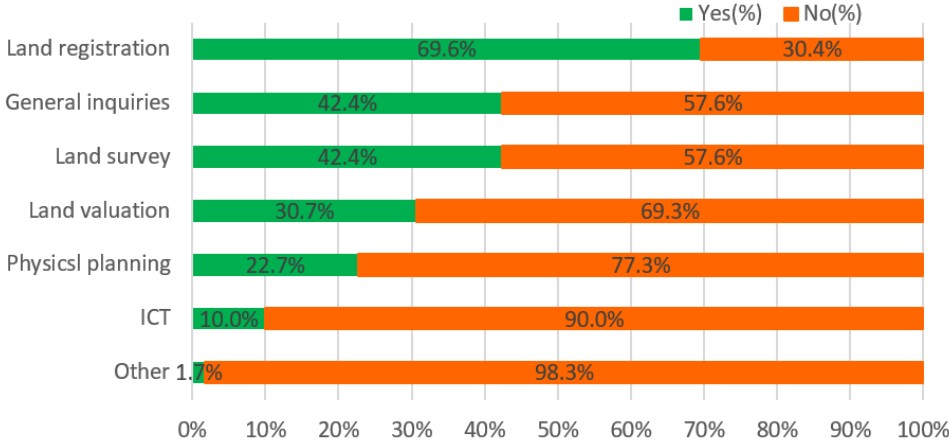

**Figure 8.** Ardhisasa platform service interactions.

*4.3. Descriptive Statistics of the Level of Satisfaction of the Current LAS*

The level of satisfaction of the participants with the current land administration system as measured by a 5-point Likert scale ranging from very dissatisfied to very satisfied was 9.0% (36), 29.9% (120), 38.9% (156), 18.7% (75), and 3.5% (14) of the respondents who responded very dissatisfied, dissatisfied, neutral, satisfied and very satisfied, respectively (see Figure 9).

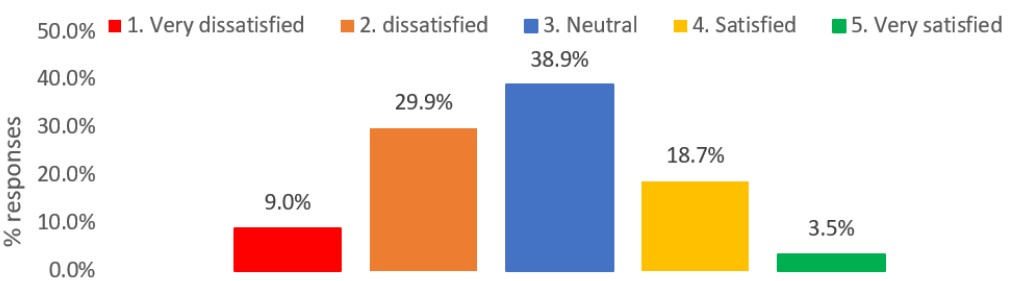

Level of satisfaction of current LAS

**Figure 9.** Descriptive statistics of the level of satisfaction.

Customer Satisfaction Score (CSAT) Index on Satisfaction with the Current LAS

The Customer Satisfaction Score (CSAT) is applied to obtain an overview of satisfied customers who have confidence in a business process [87], which can also measure LAS process outputs. CSAT is determined by asking customers to rate their satisfaction with the current LAS interaction on a five-point scale ranging from 1 (very dissatisfied) to 5 (very satisfied).

To compute the Customer satisfaction score, we targeted happy customers who were at least satisfied, which is given as follows:

$$\text{Customer satifaction score (CSAT)} = \frac{\text{satisfied} + \text{very satisfied ustomers}}{\text{number of total responses}} \times 100 \quad (3)$$

as indicated in equation three, we have (((The summation of satisfied and very satisfied responses) ÷ (Number of total responses)) × 100), which represents the percentage of satisfied customers. The Likert scale comparison for CSAT include 0%–≤20% very dissatisfied, 20%–≤40% dissatisfied, 40%–≤60% neutral, 60%–≤80% satisfied and 80%–≤100% s very satisfied. The frequency tabulation of the responses shown in Table 3 calculation yielded 22.2%.

$$\text{CSAT} = \frac{75 + 14}{401} = 22.2\%$$

**Table 3.** Level of satisfaction with the current LAS.

|       |   | Frequency | Percent | Valid Percent | Cumulative Percent |
|-------|---|-----------|---------|---------------|--------------------|
| Valid | 1 | 36        | 9.0     | 9.0           | 9.0                |
|       | 2 | 120       | 29.9    | 29.9          | 38.9               |
|       | 3 | 156       | 38.9    | 38.9          | 77.8               |
|       | 4 | 75        | 18.7    | 18.7          | 96.5               |
|       | 5 | 14        | 3.5     | 3.5           | 100.0              |
|       |   | 401       | 100.0   | 100.0         |                    |

When calculating the satisfaction rating of Table 3 based on CSAT satisfaction ratings, many customers indicated a 22.2% rating of not being satisfied.

*4.4. Comparison of Land Ownerships and Gender (n = 401)*

4.4.1. Comparison of Land Ownerships and Gender

The Pearson correlation of land ownership with gender shows a negative correlation of −0.184. Of the respondents who reported owning land, 49 (12.2%) were women, and 220 (54.8%) were men. The land ownership statistics indicate an increase in land ownership compared to the 2014 World Bank data, which stated that only 3.4% of women in Kenya owned land due to the increase in land ownership due to the new land laws that empower women and marginalised groups.

### 4.4.2. Comparison of Inquiries Made with Knowledge of Transactions

A total 80.2% of respondents who inquired about obtaining property documents knew about transactions, while 19.8% had no idea what types of documents were required. Similarly, 44.1%, 40.5%, 28.8%, 16.2%, 13.5%, and 5.4% of the respondents said they knew about the transactions, while 19.8%, 55.9%, 59.5%, 71.2%, 83.8%, 86.5%, and 94.6% had no idea about the type of documents required (see Figure 10).

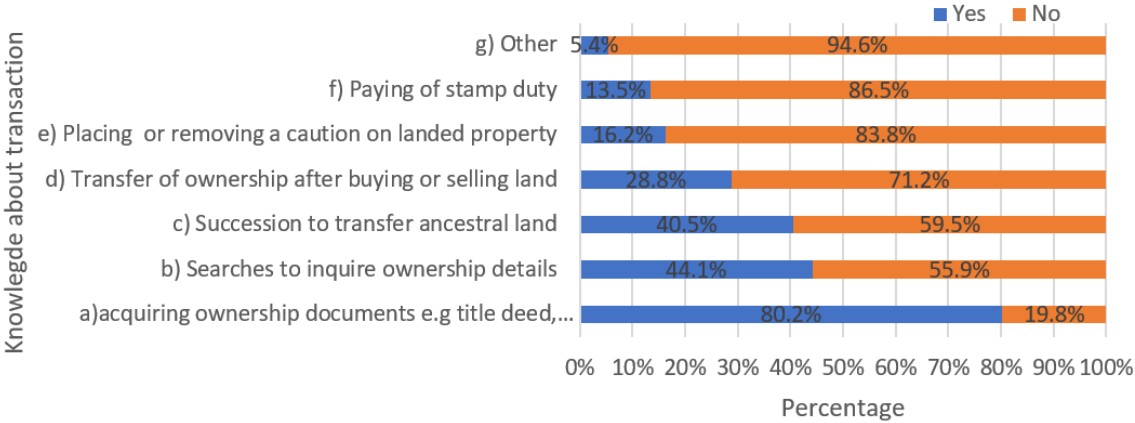

**Figure 10.** Comparison of inquiries and awareness of carrying out transactions.

### 4.4.3. Results from Interview and Focus Group Discussion

A total of forty-eight (48) respondents participated in this survey, mainly from professions in the real estate sector (see Table 4). Table A4. The composition of the focused group discussion (*n* = 48) is as follows: males were 85.4% and females were 14.6%. The age of the focused group discussion comprised age ranges of 18–29, 30–39, 40–49, and 50–59, which accounted for 45.5%, 30.3%, 18.2%, and 6.1%, respectively, with no respondent over the age of 65 years. They were selected on the premise that they have more information due to their specialisation and engagements in management, consultancy, and property management. Their task was to check the data obtained from the main responses of the Land Ministry's customers for statistical usability and to supplement their responses with regard to the effectiveness of LAS, the descriptive information on maps or plans, the most frequently requested services and the satisfaction level of LAS. The exercise lasted three days.

**Table 4.** Focussed Group discussion result on the effectiveness of LAS of Kenya (*n* = 48).

| Measurement Scale/Attribute | Not at All Effective 1 | Not So Effective 2 | Somewhat Effective 3 | Very Effective 4 | Extremely Effective 5 |
|---|---|---|---|---|---|
| Effectiveness of descriptive information of location such as district, location, registration section, parcel number, and date on Preliminary index diagram | 1 (2.1%) | 9 (18.8%) | 25 (52.1%) | 6 (12.5%) | 7 (14.6%) |
| Effectiveness of descriptive information of location such as coordinates, registration details, and parcel boundary drawing with angular and metric measurements | 1 (2.1%) | 7 (14.6%) | 21 (43.7%) | 9 (18.8%) | 10 (20.8%) |

The LAS analysis of the group discussion revealed that maps or documents alone are not sufficient to solve the problems of land registration. The percentage of those who affirmed the effectiveness of maps was 28.1% (see Table 4), while for documents, the effectiveness of satisfaction was 48.2% based on Likert scale satisfaction scores.

In terms of satisfaction level, participants in the group discussion who answered very dissatisfied, not satisfied, neutral, satisfied, and very satisfied accounted for 10.4% (5), 2.9% (11), 37.5% (18), 18.8% (9), and 10.4% (5), respectively. The results agree well with the

main findings of the 401 respondents (with mean, $\bar{x}$ = 2.75; standard error of the mean; $\sigma_M$ = 0.048; standard deviation, $\sigma$ = 0.969, for the 401 respondents) as the CSAT index is 22.2%. Wherein this case of group discussion, the CSAT index = $(5 + 9)/48 \times 100$ = 29.1% ($\bar{x}$ = 2.875, $\sigma$ = 1.111), which falls within the range of not satisfied similarly to the main statistics score, which produced a value of 22.2% and $\sigma$ = 0.969. The slight difference can be attributed to using the small sample size of better-informed individuals.

*4.5. Discussion*

Comparison of the survey with the respondents' history of dealing with registration or transfer of land and knowledge of the process they go through in registering land revealed that most of the respondents who inquired about the acquisition of property were dealing with registration or transfer of land. The nature of the requests was compared with respondents who had not made a transaction and were not aware of the process (28%), and those who had made a transaction and were aware of the process (72%).

In this research, the authors focus on the internal process of the Integrated Performance Measurement System (IPMS) framework [44] developed to assess the performance of LAS of Kenya, Ministry of Lands and Physical Planning. Based on this framework, the questions were formulated to measure the impact of the Ardhisasa' NLIMS platform on the operational process of land administration in terms of services offered to the public, including problems handled and overall customer satisfaction. To this end, fourteen open-ended and cross-ended questions were included (see Appendix A). The survey of key information included interviews with the clerks who received the documents, the customers of the Land Department, some selected key employees of the Department, and anonymous respondents, for a total of 401 respondents. The research survey analyses the characteristics of the documents that customer feedback indicates contribute the most and the least.

The assessment of Kenya LAS revealed that great efforts had been made there to draft and enact numerous laws to address the individual concerns raised by stakeholders or interest groups as noted in the current LAS. It is anticipated that more issues will arise in recording, maintenance, continuous updating, and expansion of registered lands. Improvements have been made to laws governing the registration, amendment, and deletion of security interests in Kenya's collateral registry, specifically on real estate, which can also be registered online [88]. The assessment process involves evaluating the number of procedures, the number of days and the costs of completing a procedure in the authorities.

The potential risks depend on the registration procedure used and the documentation submitted. All the processes and transactions involving many stakeholders require one to conduct due diligence of analysing real estate property registers before selling or buying the properties [11]. Due diligence is necessary for the avoidance of risk and to obtain advice from qualified personnel. Some of the procedures assessed require clients to have some prior knowledge of the documentation required, the applicable laws, or the technical details provided in the documents submitted to reduce delays in processing. For example, 80.2% of respondents who inquired about obtaining property documents knew about transactions, while 19.8% had no idea what documents were required. The same is true for searches, probate, and conveyancing, with some variation across services, as indicated in Figure 10. Regular joint training sessions for staff and clients [12] on agreed days such as the open day each financial year are recommended, in addition to continuous evaluations and monitoring. The Lands Ministry and other stakeholders could supplement training with informative help functions provided by the online Ardhisasa platform after extending the services offered in the platform as indicated by the survey and increasing access to ownership and transaction process information. The results show that land ownership by women increased from 3.4% [72] to 12.2%, as per study statistics. Then, increased ownership indicates that there are some improvements towards empowering women in taking up land-ownership initiatives with legislation protecting and supporting them in owning land as property.

Regarding the usage of paper maps and survey plans for cadastral purposes, the results indicated minimal usage of topographic maps (19.7%) and other thematic maps (8%) as compared to survey plans (52.6%), while searching ownership data accounting for 77.3% of the respondents (see Figure 5). The declining use of printed maps may be attributed to accessing up-to-date and free OpenStreetMap, ESRI, and Google Map services. In addition, image support services such as Plexi earth [89] are easily available over the internet. The Image support aid in searching, locating, and overlaying cadastral and image data on their service with provisions of using either mobile devices or desktop computers compared to outdated paper maps, which are neither provided online nor are free to use access or use.

*4.6. The Limitations of the Study*

In-person interaction with all respondents might have provided further details and information on the form. Still, respondents were given the online or alternative printed form and asked to complete it themselves. The use of printed documents and their access limited the amount of information collected from participants. Suggested solutions to the problems in land administration ranged from the introduction of the use of VGI data, legislative changes such as the introduction of new statutes and laws through legal notices, and the use of force by some actors to regain or acquire property rights. Respondents also shared limited information on land issues between county governments in Kenya and the national government, which were not included in the study. Questions 11 and 12 of the questionnaires were not included in the main survey and were only shared in the focused group discussions.

**5. Conclusions**

The first goal of this research was to find out what stakeholders or actors in land administration and management look for as information on property registration and ownership transfers. It was established that most clients who sought services at land offices responded to the survey claiming to know the entire land registration process or transfer, with 85.8% knowing while 14.2% did not. According to the hypothesis, satisfaction performance feedback received a rating of 22.2% (89/401) based on the number of clients who responded positively. Individuals who know the process mainly account for 85.8% of the total and 14.2% do not. There is a need to breach the gap by reducing the number of procedures, days, and costs associated with transactions in registration processes that can be improved to reflect changes in legislation affecting land use and administration. Improvement of LAS on procedures can be made by first providing most of the services in the online Ardhisasa-NLIMS portal in a one-stop shop, making transactions and access to information and increasing awareness of processing and registration procedures, thus making access to ownership data easier; secondly, by the integration of the register with the georeferenced parcel and geographical names service by incorporating crowdsourced information cadastral gazetteer; and thirdly, by upscaling title registration and resolving land disputes. The platform should be integrated with gazetteers and use fit-for-purpose methods to enhance current LAS in faster processing and verification for addressing concerns mentioned in land registration.

The study also aimed to determine the causes of registration problems and emerging issues in land administration after enacting new laws and mitigating the possibility of using VGI data approaches to supplement publicly available data attributes, thus hastening decision making. To rule out a lack of information as the primary cause of land issues and that authoritative data are detailed for selective areas, stakeholders and the Lands Department should organise regular joint training to identify areas of priority jointly. The study's hypothetical setup and background relied on the premise that a lack of required land attributes for registering land documents at various stages of land registration shapes people's experiences on how they can have their land registered at land offices. Using a basic framework for measuring the registration processes, it was evident that improvements on verification systems can mitigate most delays on registration, obtain internal controls

used to analyse submitted data, and perform informed internal and external inquiries. Furthermore, national mapping agencies should resolve obstacles brought up by newly enacted land laws to sustain and increase awareness of land policy concerns, particularly for clients inexperienced with real estate transactions. In conclusion, LASs should be at the forefront of leveraging services with technology and integrating cadastral gazetteers with registry information of interests and parcel locations. More studies are needed to forecast the criteria for monitoring the association between increased urban land-administration activities as a disadvantage and decreasing agricultural and natural areas for sustainable LAS.

**Author Contributions:** Conceptualisation, D.O.N.; methodology, D.O.N.; software, D.O.N.; validation, D.O.N.; formal analysis, D.O.N.; investigation, D.O.N.; resources, D.O.N.; data curation, D.O.N.; writing—original draft preparation, D.O.N.; writing—review and editing, D.O.N. and M.G.; visualisation; supervision, M.G. All authors have read and agreed to the published version of the manuscript.

**Funding:** This research received no external funding.

**Institutional Review Board Statement:** Not applicable.

**Informed Consent Statement:** Not applicable.

**Data Availability Statement:** All datasets used in this study are available from the corresponding author upon request.

**Acknowledgments:** The author acknowledges the editor and the reviewers and their critical review that helped improve the paper. The authors would also like to thank Gábor Gercsák for proofreading the article. The results presented are part of a research project for postgraduate studies. The authors would also like to acknowledge the support of ELTE Eötvös Loránd University, Faculty of Informatics, Institute of Cartography and Geoinformatics, for administrative support. The support of the Ministry of Lands is also acknowledged for providing data and allowing the performance evaluation to be undertaken. We worked under Kenya Research Permit NACOSTI/P/21/13196.

**Conflicts of Interest:** The authors declare no conflict of interest.

## Appendix A. Demographic of Questionnaire Participants

**Table A1.** Demographic of questionnaire participants.

|  | Frequency | Percent | Valid Present | Cumulative Percent |
|---|---|---|---|---|
| Male | 306 | 76.3 | 76.3 | 76.3 |
| Female | 95 | 23.7 | 23.7 | 100.0 |
| Total | 401 | 100.0 | 100.0 | |

**Table A2.** Age of Respondents.

|  | Frequency | Percent | Valid Present | Cumulative Percent |
|---|---|---|---|---|
| 18–29 | 152 | 37.9 | 37.9 | 37.9 |
| 30–39 | 161 | 40.1 | 40.1 | 78.1 |
| 40–49 | 66 | 16.5 | 16.5 | 94.5 |
| 50–59 | 21 | 5.2 | 5.2 | 99.8 |
| Over 60 | 1 | 0.2 | 0.2 | 100.0 |
| Total | 401 | 100.0 | 100.0 | |

**Table A3.** The profession of respondents (*n* = 401).

| Profession | Frequency | Percent | Valid Present | Cumulative Percent |
|---|---|---|---|---|
| Accountant | 10 | 2.5 | 2.5 | 2.5 |
| Chemist | 2 | 0.5 | 0.5 | 3.0 |
| Community Health Officer | 2 | 0.5 | 0.5 | 3.5 |
| Computer scientist | 1 | 0.5 | 5.2 | 3.7 |
| Data scientist | 1 | 0.5 | 0.5 | 4.0 |
| Developer | 1 | 0.5 | 0.5 | 4.2 |
| Director | 1 | 0.5 | 0.5 | 4.5 |
| Medical Doctor | 3 | 0.7 | 0.7 | 5.2 |
| Economist | 8 | 2.0 | 2.0 | 7.2 |
| Engineer | 55 | 13.7 | 13.7 | 20.9 |
| Environmentalist | 6 | 1.5 | 1.5 | 22.4 |
| Farmer | 2 | 0.5 | 0.5 | 22.9 |
| Fast Moving Consumer Goods | 1 | 0.2 | 0.2 | 23.2 |
| Geologist | 2 | 0.5 | 0.5 | 23.7 |
| GIS Officer | 11 | 2.7 | 2.7 | 26.4 |
| Health care worker | 1 | 0.2 | 0.2 | 26.7 |
| Human Resource officer | 4 | 1.0 | 1.0 | 27.7 |
| Insurance Underwriter | 1 | 0.2 | 0.2 | 27.9 |
| Interior Designer | 2 | 0.5 | 0.5 | 28.4 |
| ICT Officer | 14 | 3.5 | 3.5 | 31.9 |
| Advocate | 1 | 0.2 | 0.2 | 32.2 |
| Lab Analyst | 1 | 0.2 | 0.2 | 32.4 |
| Lands Officer | 4 | 1.0 | 1.0 | 33.4 |
| Land Surveyor | 113 | 28.2 | 28.2 | 61.6 |
| Lawyer | 14 | 3.5 | 3.5 | 65.1 |
| Lecturer | 9 | 2.2 | 2.2 | 67.3 |
| Marketing manager | 3 | 0.7 | 0.7 | 68.1 |
| Nurse | 10 | 2.5 | 2.5 | 70.6 |
| Operations and Expansions manager | 1 | 0.2 | 0.2 | 70.8 |
| Paramedic | 1 | 0.2 | 0.2 | 71.1 |
| Pharmacist | 1 | 0.2 | 0.2 | 71.3 |
| Agribusiness Specialist | 1 | 0.2 | 0.2 | 71.6 |
| Photogrammetrist | 2 | 0.5 | 0.5 | 72.1 |
| Physical planner | 2 | 0.5 | 0.5 | 72.6 |
| Political scientist | 1 | 0.2 | 0.2 | 72.8 |
| Procurement officer | 1 | 0.2 | 0.2 | 73.1 |
| Project manager | 1 | 0.2 | 0.2 | 73.3 |
| Public administrator | 5 | 1.2 | 1.2 | 74.6 |
| Quantity Surveyor | 1 | 0.2 | 0.2 | 74.8 |
| Real Estate Developer | 7 | 1.7 | 1.7 | 76.6 |
| Researcher | 4 | 1.0 | 1.0 | 77.6 |
| Architect | 1 | 0.2 | 0.2 | 77.8 |
| Sales executive | 1 | 0.2 | 0.2 | 78.1 |
| Security | 1 | 0.2 | 0.2 | 78.3 |
| Social worker | 1 | 0.2 | 0.2 | 78.6 |
| Software Engineer | 3 | 0.7 | 0.7 | 79.3 |
| Statistician | 1 | 0.2 | 0.2 | 79.6 |
| Student | 2 | 0.5 | 0.5 | 80.0 |
| Supply chain manager | 1 | 0.2 | 0.2 | 80.3 |
| System Security Officer | 2 | 0.5 | 0.5 | 80.8 |
| Teacher | 43 | 10.7 | 10.7 | 91.5 |
| Technologist/ Technician | 3 | 0.7 | 0.7 | 92.3 |
| Banker | 3 | 0.7 | 0.7 | 93.0 |
| Tourism Officer | 1 | 0.2 | 0.2 | 93.3 |
| Valuer | 1 | 0.2 | 0.2 | 93.5 |
| Biomedical scientist | 1 | 0.2 | 0.2 | 93.8 |
| Businessperson | 5 | 1.2 | 1.2 | 95.0 |
| Cartographer | 20 | 5.0 | 5.0 | 100.0 |
| Total | 401 | 100.0 | 100.0 | |

**Table A4.** Composition of the focused group discussion (*n* = 48).

|  | Frequency | Percent | Valid Percent | Cumulative Percent |
|---|---|---|---|---|
| Accountant | 2 | 4.2 | 4.2 | 4.2 |
| Engineer | 12 | 25.0 | 25.0 | 29.2 |
| Environmentalist | 2 | 4.2 | 4.2 | 33.4 |
| ICT Officer | 1 | 4.2 | 4.2 | 37.5 |
| Lands Officer | 1 | 2.1 | 2.1 | 39.6 |
| Land Surveyor | 6 | 12.5 | 12.5 | 52.1 |
| Lawyer | 1 | 2.1 | 2.1 | 54.2 |
| Lecturer | 5 | 10.4 | 10.4 | 64.6 |
| Social worker | 1 | 2.1 | 2.1 | 66.7 |
| Statistician | 1 | 2.1 | 2.1 | 68.8 |
| Teacher | 7 | 14.6 | 14.6 | 83.4 |
| Valuer | 1 | 2.1 | 2.1 | 85.5 |
| Businessperson | 4 | 8.3 | 8.3 | 93.8 |
| Public Administrator | 1 | 2.1 | 2.1 | 95.9 |
| Community Development Officer | 1 | 2.1 | 2.1 | 98.0 |
| Cartographer | 1 | 2.1 | 2.1 | 100.0 |
| Total | 48 | 100.0 | 100.0 |  |

**Table A5.** KMO and Bartlett's Test.

| Kaiser–Meyer–Olkin Measure of Sampling Adequacy. |  | 0.900 |
|---|---|---|
| Bartlett's Test of Sphericity | Approx. Chi-Square | 6803.891 |
|  | df | 1485 |
|  | Sig. | 0.000 |

**Appendix B. Questionnaire Items**

**Age**
1. 18–29; 2. 30–39; 3. 40–49; 4. 50–59; 5. over 60

**Gender**
☐ Male ☐ Female

**1. Which profession do you identify with?**

**2. Do you own any land as a property?**
☐ Yes ☐ No

**3. Have you ever had a transaction involving registration or transfer of land ownership in landed property?**
☐ Yes ☐ No

**4. Are you aware of the process of transferring land in Kenya from one ownership to another?**
☐ Yes ☐ No

**5. What kind of inquiries have you ever made in any ministry of lands office, when registering or transferring land ownerships?**
☐ acquiring ownership documents, e.g., title deed, deed plan, survey plan, topo map or mutation
☐ searches to inquire ownership details
☐ succession to transfer ancestral land
☐ transfer of ownership after buying or selling land
☐ placing or removing a caution on landed property
☐ Paying of stamp duty
☐ other

**6. While making the inquiries mentioned previously, what kind of details were you interested in?**
☐ location of a plot or land of interest such as beacons, coordinates and or area place name
☐ area of a plot or land of interest
☐ ownership details of a plot or land

☐ general inquiry on a plot of land
☐ user status (current usage of land)
☐ demarcation of boundaries
☐ other

**7. What Kind of documentation were you asked to avail during your inquiry?**
☐ title deed
☐ deed plan, share certificate
☐ mutation
☐ spousal consent
☐ passport-size photos
☐ national Identity card
☐ transfer forms, sale agreement
☐ consent or authority to transfer
☐ share certificate
☐ KRA pin
☐ other

**8. Specifically on your visits to a land's office, what kind of schematic data attributes were you interested in?**
☐ ownership data
☐ survey plan, preliminary index diagrams
☐ topographic map
☐ control points for the parcel of interest
☐ Thematic map, e.g., tourist map, geological map, soil map
☐ mutation or scheme plan for the parcel or plot number
☐ other

**9. Land can either be owned individually, county government(s), national state corporations, individuals, and groups. What kind of land tenure ownership of land do you enjoy currently? Check all that apply.**
☐ sole ownership or groups as private ownership
☐ tenancy or leasehold
☐ inherited though customary land
☐ squatter (though not legal)

**10. What is the most common issue(s) affecting land ownerships in Kenya? Check all that apply.**
☐ land grabbing
☐ squatting
☐ absentee landlords
☐ ethnic conflict
☐ compulsory land acquisition
☐ poor filing system-non-digital records
☐ untitled - land but still having transfer and transactions without registration based on trust
☐ double allocations
☐ land under caveat-still being sold as private land
☐ exchanged land properties involving two or more parties
☐ illegal land conversions or usages, e.g., agricultural, residential, commercial etc
☐ other

**11. How effective are the descriptive information of location such as District, location, registration section, parcel number and date on Preliminary index diagram in addressing issues noted in quiz 10 or your specific issue?**
☐ Extremely effective
☐ Very effective
☐ Somewhat effective
☐ Not so effective

☐ Not at all effective

**12. How effective are the descriptive information of location such as coordinates, registration details, parcel boundary drawing with angular and metric measurements, name of government surveyor or licensed surveyor in addressing issues noted in quiz 10 or your specific issue?**

☐ Extremely effective

☐ Very effective

☐ Somewhat effective

☐ Not so effective

☐ Not at all effective

**13. What kind of services on the ministry of lands Ardhisasa online platform or physical visits are you likely to seek or sought in the past at the land's office?**

☐ Land registration

☐ Land valuation

☐ Land survey

☐ Physical planning

☐ General inquiries

☐ ICT

☐ Other

**14. How satisfied are you with the current land administration system?**

☐ Very dissatisfied

☐ Not satisfied

☐ Neutral

☐ Satisfied

☐ Very satisfied

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
