# Peer review of "Performance Evaluation of Land Administration System (LAS) of Nairobi Metropolitan Area, Kenya"

_land, doi:10.3390/land11020203_

Round 1
Reviewer 1 Report
The research focuses on the assessment of the performance of LAS in Kenya, to provide future guidelines for improvements of LAS procedures. Based on the developed framework for the assessment, the questions were formulated in an online survey to measure the impact of the LAS, the operational process of land administration in terms of services offered to the public including problems handled and overall customer satisfaction.
Research goals, methods and results of the research are clearly described. The paper is well written and well structured. The authors may emphasize whether there are similar researches for the evaluation of LAS in Kenya and what are the differences with this approach.
There are a few typos, additional blanks spaces or no spaces, that need to be corrected. Line 588 – capital letter is missing. Line 683 – missing parentheses
Author Response
Response to Reviewer 1 Comments
The research focuses on the assessment of the performance of LAS in Kenya, to provide future guidelines for improvements of LAS procedures. Based on the developed framework for the assessment, the questions were formulated in an online survey to measure the impact of the LAS, the operational process of land administration in terms of services offered to the public including problems handled and overall customer satisfaction.
Research goals, methods and results of the research are clearly described. The paper is well written and well structured. The authors may emphasize whether there are similar researches for the evaluation of LAS in Kenya and what are the differences with this approach.
Point 1: There are a few typos, additional blanks spaces or no spaces, that need to be corrected. Line 588 – capital letter is missing. Line 683 – missing parentheses
Emphasize whether there are similar researches for the evaluation of LAS in Kenya and what are the differences with this approach
Point 1: There are a few typos, additional blanks spaces or no spaces, that need to be corrected. Line 588 – capital letter is missing. Line 683 – missing parentheses
Response 1: Typos were addressed.
Correction of typos, blank spaces in line 588 were effectively addressed.
In addition, the missing parenthesis then at line 683 has been restored.
In addition, at least six(6) similar kinds of research were reviewed and included as recommended and they include:
- Ref. no. 32. Mwangi W., “Performance Evaluation of Land Administration Systems in Kenya.” Nairobi, 2008. http://erepository.uonbi.ac.ke/handle/11295/15991.
- Ref. no. 19. Kameri-Mbote P., “Kenya Land Governance Assessment Report,” Nairobi, Jun. 2016. Accessed: Jan. 13, 2022. [Online]. Available: http://documents.worldbank.org/curated/en/829991504864783043/pdf/119619-WP-P095390-PUBLIC-7-9-2017-10-9-20-KenyaFinalReport.pdf.
- Ref. no. 34. Wayumba R., “Developing Land Information Management Systems for County Governments in Kenya,” IOSR Journal of Engineering, vol. 07, no. 05, pp. 42–49, 2017, doi: 10.9790/3021-0705014249.
- Ref no. 33. Siriba, D. N. W. Voß, and. Mulaku G. C. M, “The Kenyan Cadastre and Modern Land Administration,” Zeitschrift für Vermessung, vol. 136, no. 3, pp. 177–186, 2011.
- Ref. no. 35. Wayumba G. O., “An Evaluation of the Cadastral System in Kenya and a Strategy for its Modernization,” Nairobi, 2013. Accessed: Jan. 13, 2022. [Online]. Available: http://erepository.uonbi.ac.ke/bitstream/handle/11295/56366/Wayumba,Gordon?sequence=3.
- Ref. no. 36. Koeva M., et al., “Innovative remote sensing methodologies for Kenyan land tenure mapping,” Remote Sensing, vol. 12, no. 2, pp. 1–27, Jan. 2020, doi: 10.3390/rs12020273.
The present study is different in that, there is no study on performance evaluation of the LAS in Kenya, specifically focused on the internal processes of the Kenya Land Administration. Secondly, even though the studies were done under the new land laws dispensation, issues noted for each study are different as reflected in their findings and recommendations.
Reviewer 2 Report
Dear Authors,
let me start by saying that I really like your publication.
It is on topic; it has a scientific element and, above all, it can interest an international reader. The problem of quality of cadastral data and land ownership is important and the sign that 28% of clients do not know the land registration transactions.
1.Although the article deals with Kenya, I would like to encourage/ask you to expand the part of the introduction, especially by extending the list, the cases of countries where the cadastral system is developed.
I think that in line 34-35 it is worth adding not one but the extentend list of countries based on which property information system can be developed in Kenya.
Below are a few: EU countries, Germany, Poland and Czech Republic:
a.Comparetti, A., & Raimondi, S. (2019). CADASTRAL MODELS IN EU MEMBER STATES. EQA - International Journal of Environmental Quality, 33, 55–78. https://doi.org/10.6092/issn.2281-4485/8558
b.Stoter J., Salzmann M.,(2003),Towards a 3D cadastre: where do cadastral needs and technical possibilities meet?, Computers, Environment and Urban Systems, Volume 27, Issue 4, 2003, Pages 395-410, ISSN 0198-9715,
https://doi.org/10.1016/S0198-9715(02)00039-X.
c. Jasinska, E. Real estate due diligence on the example of the polish market. In Proceedings of the 14th International Multidisciplinary Scientific Geoconference (SGEM), Albena, Bulgaria, 17–26 June 2014; Volume 2, pp. 419–426.
d.Lojka, B.; Teutscherová, N.; Chládová, A.; Kala, L.; Szabó, P.; Martiník, A.; Weger, J.; Houška, J.; Červenka, J.; Kotrba, R.; Jobbiková, J.; Doležalová, H.; Snášelová, M.; Krčmářová, J.; Vávrová, K.; Králík, T.; Zavadil, T.; Lawson, G. Agroforestry in the Czech Republic: What Hampers the Comeback of a Once Traditional Land Use System? Agronomy 2022, 12, 69. https://doi.org/10.3390/agronomy12010069
2. I also believe that it is worthwhile to take advantage of the knowledge of Hungarian authors and briefly refer to their experiences.
I really appreciate the use of current data such as: "The World Bank statistics of land tenure security in 2020".
I have doubts about the editorial part of the notation:
" ............................ equation 1" (line 404, 417)
it seems to me that lines 418-420 also need to be corrected
Figure 9 and data from the survey completed on 15 November 2021 to the best part of this article!
Although I am aware that the African cadastral system needs a huge amount of work- this is the first time I have come across such a factual approach to the subject- congratulations
4.Figure 9 and data from the survey completed on 15 November 2021 to the best part of this article!
Although I am aware that the African cadastral system needs a huge amount of work- this is the first time I have come across such a factual approach to the subject- congratulations .
I recommend publication in its present form, possibly with an expansion of the literature to include examples of developed cadastral systems in selected countries.
Author Response
Response to Reviewer 2 Comments
Dear Authors,
let me start by saying that I really like your publication.
It is on topic; it has a scientific element and, above all, it can interest an international reader. The problem of quality of cadastral data and land ownership is important and the sign that 28% of clients do not know the land registration transactions.
Point 1. Although the article deals with Kenya, I would like to encourage/ask you to expand the part of the introduction, especially by extending the list, the cases of countries where the cadastral system is developed.
Response 1:The introduction section was extended to include the following countries are recommended. Finland, France, Austria, Italy, Sweden, Luxembourg, Greece, Netherlands, Denmark, Portugal and Spain for use cases of the developed cadastral systemin addition to Germany, Poland and Czech Republic (line 157-171). In addition, a literature review of EU gazetteers and integration with cadasters was incorporated to indicate the status of integration of cadastres with gazetteers as integrated LAS system for developed countries( (line 40-147).
Point 2. I think that in line 34-35 it is worth adding not one but the extended list of countries based on which property information system can be developed in Kenya.
Below are a few: EU countries, Germany, Poland and Czech Republic:
a.Comparetti, A., & Raimondi, S. (2019). CADASTRAL MODELS IN EU MEMBER STATES. EQA - International Journal of Environmental Quality, 33, 55–78. https://doi.org/10.6092/issn.2281-4485/8558
b.Stoter J., Salzmann M.,(2003),Towards a 3D cadastre: where do cadastral needs and technical possibilities meet?, Computers, Environment and Urban Systems, Volume 27, Issue 4, 2003, Pages 395-410, ISSN 0198-9715,
https://doi.org/10.1016/S0198-9715(02)00039-X.
- Jasinska, E. Real estate due diligence on the example of the polish market. In Proceedings of the 14th International Multidisciplinary Scientific Geoconference (SGEM), Albena, Bulgaria, 17–26 June 2014; Volume 2, pp. 419–426.
d.Lojka, B.; Teutscherová, N.; Chládová, A.; Kala, L.; Szabó, P.; Martiník, A.; Weger, J.; Houška, J.; Červenka, J.; Kotrba, R.; Jobbiková, J.; Doležalová, H.; Snášelová, M.; Krčmářová, J.; Vávrová, K.; Králík, T.; Zavadil, T.; Lawson, G. Agroforestry in the Czech Republic: What Hampers the Comeback of a Once Traditional Land Use System? Agronomy 2022, 12, 69. https://doi.org/10.3390/agronomy12010069
Response 2: All the shared four references were reviewed and incorporated in the literature review as indicated by Reference no. 10, 11, 12 and 13 respectively.
Point 2. I also believe that it is worthwhile to take advantage of the knowledge of Hungarian authors and briefly refer to their experiences.
Response 2: As per the recommendation, The Hungarian portal for the cadastral system for Hungary and in addition, two authors with research experience of Hungarian cadastral system were included in the literature review and indicated by reference no.16, 17 and 18 for the Hungarian cadastral website and the two authors from Hungary.
- Ref no.19 The Government of Hungary- Ministry of Agriculture Department of Land Administration, “Official Portal of the Hungarian Land Administration.” http://en.foldhivatal.hu/content/view/46/58/ (accessed Jan. 12, 2022).
- Ref.no. 17 A. Molnár and V. Parsova, “Cadastral system in Hungary,” in Scientific Methodical Conference, Baltic Surveying, 17, 2017, pp. 17–20. Accessed: Jan. 12, 2022. [Online]. Available: http://llufb.llu.lv/conference/Baltic-surveying/2017/Baltic_surv_proceedings_2017.pdf
- Ref. no. 18 György Domokos, “Brief Overview on Hungarian Land Hungarian Land Administration Administration,” in Hungarian Geodetic Surveying Liaison Group, 2012, pp. 1–18. Accessed: Jan. 12, 2022. [Online]. Available: https://www.clge.eu/wp-content/uploads/2012/05/hungary.pdf
I really appreciate the use of current data such as: "The World Bank statistics of land tenure security in 2020".
Point 3: I have doubts about the editorial part of the notation:
" ............................ equation 1" (line 404, 417)
it seems to me that lines 418-420 also need to be corrected
Response 3: Equations 1 and 2 have been modified as per the journal requirements of writing mathematical formulas for equation 1 (line 1044-1057) and for equation 2 (line 1059-1072).
Point 4 Figure 9 and data from the survey completed on 15 November 2021 to the best part of this article!
Although I am aware that the African cadastral system needs a huge amount of work- this is the first time I have come across such a factual approach to the subject- congratulations
I recommend publication in its present form, possibly with an expansion of the literature to include examples of developed cadastral systems in selected countries
Response 4: Added five references shared by reviewer in addition to reviews for developed cadastral systems and addressing gazetteers for countries of Belgium, Czechia Republic, Spain, France, Luxemburg, Netherlands, Norway, Italy and Hungary and indicated in reference 4. 5,6, 14 and 18.
- Ref. no. 4 Cetl V., Vrečar S.,. Reuter H. I, Boguslawski R., and Pignatelli F., “EU gazetteer evaluation, EUR 30378 EN, Publications Office of the European Union, Luxembourg, JRC121541,” 2020. doi: 10.2760/095025.
2 Ref no. 5 Landgate, “Access the Cadastre (Polygons) dataset for Western Australia.” https://www.wa.gov.au/service/natural-resources/land-use-management/access-the-cadastre-polygons-dataset (accessed Jan. 12, 2022).
- Ref. no. 6 Dalrymple K., Williamson I., and Wallace J., “Cadastral Systems within Australia,” Australian Surveyor, vol. 48, no. 1, pp. 37–49, Jun. 2003, doi: 10.1080/00050357.2003.10558851.
- Ref. no. 14 The Italian Republic-Directorate for Cadastral cartographic and Land Registration Services, “The Italian Cadastral System.” Accessed: Jan. 12, 2022. [Online]. Available: https://www.agenziaentrate.gov.it/portale/documents/180690/1185444/the+italian+cadestral+system+2018_The+Italian+Cadastral+System+2018.pdf
- Ref. no. 18 György D., “Brief Overview on Hungarian Land Hungarian Land Administration Administration,” in Hungarian Geodetic Surveying Liaison Group, 2012, pp. 1–18. Accessed: Jan. 12, 2022. [Online]. Available: https://www.clge.eu/wp-content/uploads/2012/05/hungary.pdf

Reviewer 3 Report
The article is well structured, the introduction is extensive, the case study and the issues addressed are current.
The references are current and in the topic with the approached issue, the authors being up to date with the articles written in the field lately.
The methods used, the algorithms and the study are highly scientific.
The authors investigated the current issue of administration and it is well structured and current.
I propose that the article be accepted for publication.
Author Response
Response to Reviewer 3 Comments
The article is well structured, the introduction is extensive, the case study and the issues addressed are current.
The references are current and in the topic with the approached issue, the authors being up to date with the articles written in the field lately.
The methods used, the algorithms and the study are highly scientific.
The authors investigated the current issue of administration and it is well structured and current.
I propose that the article be accepted for publication.
Response 1: Thank you for your comments.

Reviewer 4 Report
The first part of the manuscript (Abstract and Section 1) is difficult to read and understand. For example, the opening line (no 28) reads: 'Interests in land and the registers govern proper land administration'. How can 'interests in land govern anything? Another example (93-94): 'the application of EFQM was implemented in Europe and then rolled to be applied in other areas such as Poland[19] the Netherlands'. ??? - Mentioning of UN-ECE guidelines and Kenya's land policy of 2007 and subsequent laws are fine, but why should the reader be bothered with the spread of the Torrens system?
The research identifies lack of information and suggest to rule out this lack of information through 'regular joint training' (line 785, and again 810). This could be supplemented with informative help-functions, provided through the online Ardhisasa platform, because the survey indicated, what information was needed. Authors propose 'centralizing and stabilizing the services ..' (801-05), but these proposals seem weaker related to the outcome of the survey.
Author Response
Response to Reviewer 4 Comments
The first part of the manuscript (Abstract and Section 1) is difficult to read and understand. For example, the opening line (no 28) reads: 'Interests in land and the registers govern proper land administration'. How can 'interests in land govern anything?
Response 1: Definitions have been made to clarify what interests and registers mean within the context of real estate and as used in the article to make the preceding contents clear, easy to read and understand.
Another example (93-94): 'the application of EFQM was implemented in Europe and then rolled to be applied in other areas such as Poland[19] the Netherlands'. ??? - Mentioning of UN-ECE guidelines and Kenya's land policy of 2007 and subsequent laws are fine, but why should the reader be bothered with the spread of the Torrens system?
Response 2: The mentioning of the Torens system clarifies and reiterates that wherever it was used, amendments have been sought to address issues related to its application in addition to the introduction of new land laws to supersede or replace the continued application of the deed plan system based on Torrens based laws.
The research identifies lack of information and suggest to rule out this lack of information through 'regular joint training' (line 785, and again 810). This could be supplemented with informative help-functions, provided through the online Ardhisasa platform, because the survey indicated, what information was needed. Authors propose 'centralizing and stabilizing the services ..' (801-05), but these proposals seem weaker related to the outcome of the survey
Response 3: The researchers have improved the language of expression that clarified that information on registration and ownership data is packaged in a way that makes only those who have been involved with the transactions have access to information on the awareness of the procedures of LAS, while those who have not, have difficulty in making a transaction without issues.

Reviewer 5 Report
This paper aims to investigate insights into the internal process of the current land administration of Kenya. The authors evaluated the performance of the land administration of Kenya based on the findings from the questionnaire survey. I have read the article in full. While the authors tried to present the results of their work, the article is poorly organized. The structure of the article must be improved. The authors are advised to follow the well-structured articles to organize their paper. The quality of English writing must be improved-language editing is required. In most cases, the structuring of the sentences does not follow the grammar. In some cases, the length of the sentence is large which created ambiguity in the real meaning of the sentence. I suggest the author to improve the paper by restructuring the paper in a good manner (please have a look at the structure of other published papers) and presenting their result in an organized way. I also suggest the authors to consult the experienced article writer(s) to reorganize the overall structure of the paper. My specific comments are as follows. The structure of the paper and quality of English language writing must be improved to become the paper publishable.
- The title can be modified as “Performance evaluation of Land Administration System (LAS) of Nairobi Metropolitan area, Kenya”
- The article should be properly formatted as per the journal’s guidelines. Proper numbering should be used in different sections and sub-sections. The naming of sub-sections should be well descriptive. For example, a) “Land Administration System (LAS) performance evaluation” in line 92: Is it a sub-title? If yes, then please format as per guideline. Otherwise, I think it is not necessary. “The questionnaire distribution and the stakeholder survey sample size estimation” in line 400: What is it? It’s sub-section or not? “The question items” in line 430- what is this? The authors should be careful about this.
- Introduction: The introduction section is too large and not well structured. It also lacks coherence and continuity. Authors should describe, first, the problem statement, the relevant works and methods available in the literature, the methods in other works, the method they are going to use in their study, and finally their contribution. Although the authors presented most of the items, but they are not logically ordered. For example, the authors presented relevant methods in lines 56-75 and relevant works were presented in lines 93-117. However, relevant works should come first, then the description of relevant methods. The length of the introduction must be reduced by removing unnecessary texts.
- Method: The description of the method section is not well described. In the introduction section, the authors mentioned that they used Integrated Performance Measurement System (IPMS) method to evaluate the LAS (line 84), however, there is no description of IPMS in the method section. The authors should outline the IPMS method in the method section. The authors used two equations (equations 1 and 2) to determine the sample size. But did not mention the final sample size.
- Result: Although the findings were described but the presentation style is not up to the mark. The presentation style and result need to be improved and organized step by step. Although the findings of the survey have been presented, I did not find the result of IPMS although it was supposed to do that.
- The limitation should come after the discussion section
- Discussion: There connection between results and discussion is weak. In the discussion section, the authors should interpret the results in detail. What is the significance of their result?

Author Response
Response to Reviewer 5 Comments
This paper aims to investigate insights into the internal process of the current land administration of Kenya. The authors evaluated the performance of the land administration of Kenya based on the findings from the questionnaire survey. I have read the article in full. While the authors tried to present the results of their work, the article is poorly organized. The structure of the article must be improved. The authors are advised to follow the well-structured articles to organize their papers. The quality of English writing must be improved-language editing is required. In most cases, the structuring of the sentences does not follow the grammar. In some cases, the length of the sentence is large which created ambiguity in the real meaning of the sentence. I suggest the author to improve the paper by restructuring the paper in a good manner (please have a look at the structure of other published papers) and presenting their result in an organized way. I also suggest the authors to consult the experienced article writer(s) to reorganize the overall structure of the paper. My specific comments are as follows. The structure of the paper and quality of English language writing must be improved to become the paper publishable.
Point 1. The title can be modified as “Performance evaluation of Land Administration System (LAS) of Nairobi Metropolitan area, Kenya”
Response 1: Modified as requested to read “Performance evaluation of Land Administration System (LAS) of Nairobi Metropolitan area, Kenya”.
Point 2. The article should be properly formatted as per the journal’s guidelines. Proper numbering should be used in different sections and sub-sections. The naming of sub-sections should be well descriptive. For example, a) “Land Administration System (LAS) performance evaluation” in line 92: Is it a sub-title? If yes, then please format as per guideline. Otherwise, I think it is not necessary. “The questionnaire distribution and the stakeholder survey sample size estimation” in line 400: What is it? It’s sub-section or not? “The question items” in line 430- what is this? The authors should be careful about this.
Response 2: The numberings of the different sections and subsections have been done as recommended for the affected sections such as 1.1., 1.2,1.3 and 1.4 for sections 1 and 2.1, 2.2 and 2.3 for section 2 (with subsections 2.3.1, 2.3.2, 2.3.3 and 2.3.4)
Point 3. Introduction: The introduction section is too large and not well structured. It also lacks coherence and continuity. Authors should describe, first, the problem statement, the relevant works and methods available in the literature, the methods in other works, the method they are going to use in their study, and finally their contribution. Although the authors presented most of the items, but they are not logically ordered. For example, the authors presented relevant methods in lines 56-75 and relevant works were presented in lines 93-117. However, relevant works should come first, then the description of relevant methods. The length of the introduction must be reduced by removing unnecessary texts.
Response 3: The introduction section also contains the literature review hence it is not large as pointed out. However, modification has been made regarding making relevant works to be followed by a description of the relevant methods of performance evaluation[line 191-320] and also for [ine 321-22] for previous works in Kenya on LAS. All text in the introduction is relevant, however, there was a minimal retraction of some text and one reviewer asked for its expansion hence it expanded slightly. The language used in the text has been revised to improve grammar and remove ambiguity as recommended.
Point 4. Method: The description of the method section is not well described. In the introduction section, the authors mentioned that they used Integrated Performance Measurement System (IPMS) method to evaluate the LAS (line 84), however, there is no description of IPMS in the method section. The authors should outline the IPMS method in the method section. The authors used two equations (equations 1 and 2) to determine the sample size. But did not mention the final sample size.
Response 4: The method section was briefly described initially. However, currently, it has been made detailed, at introduction section(line 420-452) and in the methods section 2 especially the explanation of the integrated performance measurement (IPMS) criteria or parameters items used in the questionnaire framework used in the evaluation of the LAS under section 2, sub-section 2.3.1, 2.3.2, 2.3.3 and 2.3.4 [line 1035-1191] in the methods section subsection ‘2.3.4 Statistical evaluation of inquiries of ownership details and internal processes’(See table 2)
Regarding the two equations used; the final sample size used was reported in lines 421-‘for our case we got sufficient data from 401 respondents’, 429 –‘yields 400 as the minimum required responses, also 401 respondents participated’ in the methods sections. In the results section, (lines 507, 512,520 ad infinitum) as well as in every statistical table or figure as n=401, where n is the final population sample size used. Now it is found on lines 1071-1072.
Point 5. Result: Although the findings were described but the presentation style is not up to the mark. The presentation style and result need to be improved and organized step by step. Although the findings of the survey have been presented, I did not find the result of IPMS although it was supposed to do that.
Response 5: The results of the IPMS methodology has been described in detail as recommended (see Table 1-line 428-429), as well as how it was implemented (See Table 2-1190-1191) under section 2.3.4 has its results presented in Figure 2-9.
Point 6. The limitation should come after the discussion section.
Response 6: Limitation was moved to be located after the discussion section as recommended.
Point 7. Discussion: There connection between results and discussion is weak. In the discussion section, the authors should interpret the results in detail. What is the significance of their result?
Response 7: The discussion of results has been improved to support the descriptions presented using Figures 2-9 and Tables 1-2. In another instance, there is the use of satisfaction index parameters that make it describe the results and provide a summary discussion at the results section for all the items presented as per the goals of the present work.

Round 2
Reviewer 4 Report
The study is broad, implying an overwhelming diversity of facts, proposals, and outcomes of the survey. The conclusion may suggest next steps
Author Response
Point1:
The study is broad, implying an overwhelming diversity of facts, proposals, and outcomes of the survey. The conclusion may suggest next steps
Response 1: The manuscript's grammar was checked, refined and proofread.
Errors found were corrected.
Some steps to be undertaken are briefly “to have LASs at the forefront of leveraging services with technology and integrating cadastral gazetteers with registry information of interests and parcel locations” (see line 1891-1893).

Reviewer 5 Report
The authors tried to address most of the comments but English editing is still required. I recommend again improving the English style with the help of an experienced writer. A few examples are as follows:
Line 32: comma missing. In real property interest in land is the right to acquire and own land.
The corrected sentence may be: In real property, interest in land is the right to acquire and own land.
Line 38-41: large sentence, unclear meaning
Line 194: incorrect structure of sentence leads to unclear meaning.
The present works evaluated the internal process only of the Kenya Land Administration System (LAS), ………………
Corrected: The present works evaluated only the internal process of the Kenya Land Administration System (LAS), ……………………
Line 168-170: full stop missing
Line 195-200, 201-205, the large sentence makes it difficult to understand the proper meaning
Additional comments:
Formating needs to be carefully done.
Table numbering is not still corrected. Table 1 in two places-line 186 and 785, Table 2 in two places-line 552 and 830
Figure 1 in two places-line 455 and 635
Please correct the numbering as well as in-text reference.
Author Response
Point 1:
The authors tried to address most of the comments but English editing is still required. I recommend again improving the English style with the help of an experienced writer. A few examples are as follows:
Line 32: comma missing. In real property interest in land is the right to acquire and own land.
The corrected sentence may be: In real property, interest in land is the right to acquire and own land.
Line 38-41: large sentence, unclear meaning
Line 194: incorrect structure of sentence leads to unclear meaning.
The present works evaluated the internal process only of the Kenya Land Administration System (LAS), ………………
Corrected: The present works evaluated only the internal process of the Kenya Land Administration System (LAS), ……………………
Line 168-170: full stop missing
Line 195-200, 201-205, the large sentence makes it difficult to understand the proper meaning
Response 1: Errors in grammar, typos were corrected or refined and proofread for clarity. The long sentences were split accordingly.
The issue of captioning the figures and tables has been addressed for Tables 1-4 and Figure 1-10.
Formating of text was done for consistency. The manuscript's grammar was also checked using Grammarly software to check for any
Point 2 : Additional comments:
Formating needs to be carefully done.
Table numbering is not still corrected. Table 1 in two places-line 186 and 785, Table 2 in two places-line 552 and 830
Figure 1 in two places-line 455 and 635
Please correct the numbering as well as in-text reference.
Response 2: Formating and the numberings of the tables and figures were corrected as recommended as well as in the text’s manuscript.
In detail for tables, Table 1 is in line 503-504, Table 2 line 1188-1189, Table 3 line 1584-1585 and Table 4 line 1682-1683. Similarly for figures, Figure 1 Line 1022-1035, Figure 2 line 1317-1318, Figure 3 line 1253-1254, Figure 4 line 1403-1404, Figure 5 line 1449-1450,
